# Groundwater–CO$_2$ Emissions Relationship in Dutch Peatlands Derived by Machine Learning Using Airborne and Ground-Based Eddy Covariance Data

Laura M. van der Poel[1], Laurent V. Bataille[1], Bart Kruijt[1], Wietse Franssen[1], Wilma Jans[1], Jan Biermann[1], Anne Rietman[1], Alex J. V. Buzacott[2], Ype van der Velde[3], Ruben Boelens[4], and Ronald W. A. Hutjes[1]

[1]Earth Systems & Global Change Group, Wageningen University, 6708 PB Wageningen, Netherlands
[2]Institute for Atmospheric and Earth System Research, University of Helsinki, 00014 Helsinki, Finland
[3]Faculty of Science, Earth and Climate, Vrije Universiteit Amsterdam, 1081 HV Amsterdam, Netherlands
[4]HydroLogic, 3811 HN Amersfoort, Netherlands

**Correspondence:** Laura M. van der Poel (lvdp@ign.ku.dk) and Ronald W. A. Hutjes (ronald.hutjes@wur.nl)

**Abstract.** Peatlands worldwide have been transformed from carbon sinks to carbon sources due to years of intensive agriculture requiring low water tables. In the Netherlands, carbon dioxide (CO$_2$) emissions from drained peatlands mount up to 5.6 Mton annually and, according the Dutch climate agreement, should be reduced by 1 Mton in 2030. It is generally accepted that mitigation measures should include raising the water level, and the exact influence of water table depth has been increasingly studied in recent years. Most studies do this by comparing annual Eddy Covariance (EC) site-specific CO$_2$ budgets to mean annual effective water table depths. However, here we apply a different approach: we integrate measurements from 16 EC towers with EC measurements from 141 flights by a low-flying research aircraft, in an interpretable machine learning (ML) framework. We make use of the different strengths of tower and airborne data, temporal continuity and spatial heterogeneity, respectively. We apply time frequency wavelet analysis and a footprint model to relate the measured fluxes to the underlying surface. Using spatio-temporal data, we train and optimize a boosted regression tree (BRT) machine learning algorithm to predict immediate CO$_2$ fluxes and use Shapley values and various simulations to interpret the model's outputs. We find that emissions increase with 4.6 tonnes CO$_2$ ha$^{-1}$ yr$^{-1}$ (90% CI: 4.0-5.4) for every 10 cm lowering of the water table, down to a water table depth of 0.8 meter below the surface. For more drained conditions, emissions decrease again. Furthermore, we find that the sensitivity of CO$_2$ emissions to drainage is stronger in winter than in summer and that it varies between sites. This study shows the added value of using ML with different types of instantaneous data, and holds potential for future applications.

## 1 Introduction

Despite covering only 3% of the earth's land surface, peatlands store around 25% of all terrestrial carbon and play a crucial role in the global carbon cycle (Yu et al., 2010). They are the most carbon-dense ecosystems of the terrestrial biosphere and have a true potential for climate change mitigation (Leifeld and Menichetti, 2018; Loisel et al., 2021). In natural, waterlogged fens and bogs, uptake of carbon dioxide (CO$_2$) through vegetation and subsequent sequestration in peat soils, abundantly exceeds

the emission of methane ($CH_4$) (Frolking and Roulet, 2007). However, peat soils have been exploited and drained worldwide for fuel extraction and agricultural practices. They widely transformed from carbon sinks to carbon sources due to increased peat decomposition following higher oxygen availability, and are currently responsible of large $CO_2$ emissions.

The Netherlands has a long history of peat extraction and intensively draining peatlands for agriculture and livestock farming (van den Akker et al., 2008; Erkens et al., 2016). This has led to increased carbon dioxide emissions, currently accounting for $\sim$3% of all Dutch emissions (5.6 Mton annually), and land subsidence, which in turn increases the need for further drainage (Kwakernaak et al., 2010; Ruyssenaars et al., 2022). To counter this spiral, the Dutch government set a specific mitigation target for peat meadows: annual emissions must be reduced by 1 Mton by 2030 (Government of the Netherlands, 2019). It is generally accepted that counter measures to reduce such emissions should include raising the water table, as water table seems the predominant control on greenhouse gas emissions from managed peatlands (Evans et al., 2021). However, the exact impact of higher groundwater levels on $CO_2$ fluxes is not yet entirely established, and has been increasingly studied in the past years (Aben et al., 2024; Boonman et al., 2022; Evans et al., 2021; Fritz et al., 2017; Kruijt et al., 2023; Tiemeyer et al., 2020).

Studies investigating agricultural systems generally require correcting the annual net ecosystem exchange (NEE) for lateral movement of carbon associated with manure applications and harvests, and relate the resulting net ecosystem carbon balance (NECB) to mean annual groundwater level (Aben et al., 2024; Boonman et al., 2022; Evans et al., 2021; Kruijt et al., 2023). In unmanaged systems, annual NEE is equivalent to NECB. Averaging out daily and seasonal variation, the goal is to isolate the underlying effect of groundwater level. This way, multiple studies found linear relationships with similar slopes: between 2.1 and 4.5 t $CO_2$ ha$^{-1}$ yr$^{-1}$ extra emissions per 10 cm increase in WTDe (Boonman et al., 2022; Evans et al., 2021; Fritz et al., 2017; Jurasinski et al., 2016; Kruijt et al., 2023). Tiemeyer et al. (2020) fitted the Gompertz function to a set of annual balances, which shows a sharper increase in emissions at shallow water levels, but then saturates at around 0.4 m. Recently, multiple studies have applied this function (Friedrich et al., 2024; Koch et al., 2023; Nijman et al., 2024).

While studies on annual budgets provide valuable insights into the underlying groundwater-$CO_2$ relationship and differences between sites, some limitations emerge. First, to obtain an NECB estimate for any location, year-round data at that specific location is required, which is not always achievable and generally requires some trustworthy gap-filling. Second, whilst carbon import and export can add up to significant amounts, these numbers are often hard to obtain and generally unavailable at landscape level. Third, the site comparisons are robust only when comparing sites with markedly different average water table depths. The datasets used have been based on site specific observations, with well-defined, fairly homogeneous soil and vegetation characteristics and well-known water table management. However, these factors generally vary widely on the regional scale. Last, the annual estimates discard possibly important information from intra-annual variability and relationships with other factors than groundwater. In the current study, we aim to by-pass these limitations by alternatively exploring the short time scale at the regional level, to further unravel the influence of water table depth and other key drivers on $CO_2$ emissions from agricultural peatlands in the Netherlands.

We do this by incorporating flux measurements from a low-flying aircraft. Airborne measurements bear high spatial heterogeneity, since every measured flux originates from an area called the footprint spanning several kilometers, and an entire region can be covered by an appropriate flight pattern. However, a limitation of the airborne measurements is that they are

generally limited to daytime conditions. This limitation is particularly critical for $CO_2$ studies, given the different contributions of Gross Primary Productivity (GPP) and Ecosystem Respiration (Reco) to NEE. Here, we are specifically interested in the peat decomposition component of heterotrophic respiration, unrelated to GPP. Measurements by Eddy Covariance towers, on the other hand, are continuous and include nighttime fluxes, enabling NEE partitioning, but are limited by their fixed location. Therefore, we consider complementary use of tower and airborne flux estimates essential to assess $CO_2$ fluxes at a regional scale.

Airborne measurements (integrated flux signals from their respective footprints) can be related to the underlying surface by environmental response functions using either more classical statistical methods (Hutjes et al., 2010), or by artificial intelligence approaches (Metzger et al., 2013, 2021; Serafimovich et al., 2018). Both depend on overlaying the footprint of all flux measurements over maps of vegetation, land use and soils and/or direct satellite derived products. Here, we integrate tower and airborne data using the 'LTFM' approach initially developed by Metzger et al. (2013), which includes four principal steps: Low level flights, Time frequency wavelet analysis, Footprint modeling, and Machine learning. We aim to apply this machine learning (ML) approach to identify key predictors of NEE and understand their influence on $CO_2$ dynamics in Dutch peatlands.

Today, peat soils in the Netherlands can be found in the 'Groene Hart' and in the northwest, in the provinces of Friesland and Overijssel. These areas share similarities, such as being mostly dominated by pastures with ditches, but also differ in certain aspects, such as average drainage depth, Friesland being the most intensively drained with lowest water tables. In the current study, we have three distinct flight tracks: above the Groene Hart area, southern Friesland, and the western part of Overijssel. These three regions are also equipped with flux towers, thus we measure both airborne and tower $CO_2$ fluxes from all three.

In our aim to understand the influence of key predictors, we inspect potential differences and similarities between these areas. We do this by building several models: one model based on all data together, and models per area separately, trained only on the data of the respective area. We expect the model with all data will perform best, since in machine learning data quantity is often a determining factor for performance. Furthermore, we expect that although characteristics of the areas vary, the underlying processes do not, hence 'one model fits all', and area-specific models can be used to predict for other areas. To achieve physical interpretability of our ML-approach, we use the SHAP framework and model simulations, fully exploring the identified relationships. We hypothesize that, alongside drivers of the diurnal cycle, water table depth plays an important role at the regional scale in all areas, and that the machine learning approach can help us to better understand drivers of $CO_2$ fluxes. We aim to finally create a robust, interpretable ML model that can be used in the Netherlands to predict $CO_2$ emissions from drained fen meadows at a regional scale.

## 2 Methods

In this study, we merge airborne flux measurements with ground-based Eddy Covariance (EC) measurements, to make use of their spatial and temporal strengths, respectively. Both are part of the intensive monitoring network implemented by the Netherlands Research Programme on Greenhouse Gas Dynamics in Peatlands and Organic Soils (NOBV, https://www.nobveenweiden.nl/en/), that was established following the Dutch mitigation target of 2019. The goal of the NOBV is to further understand greenhouse

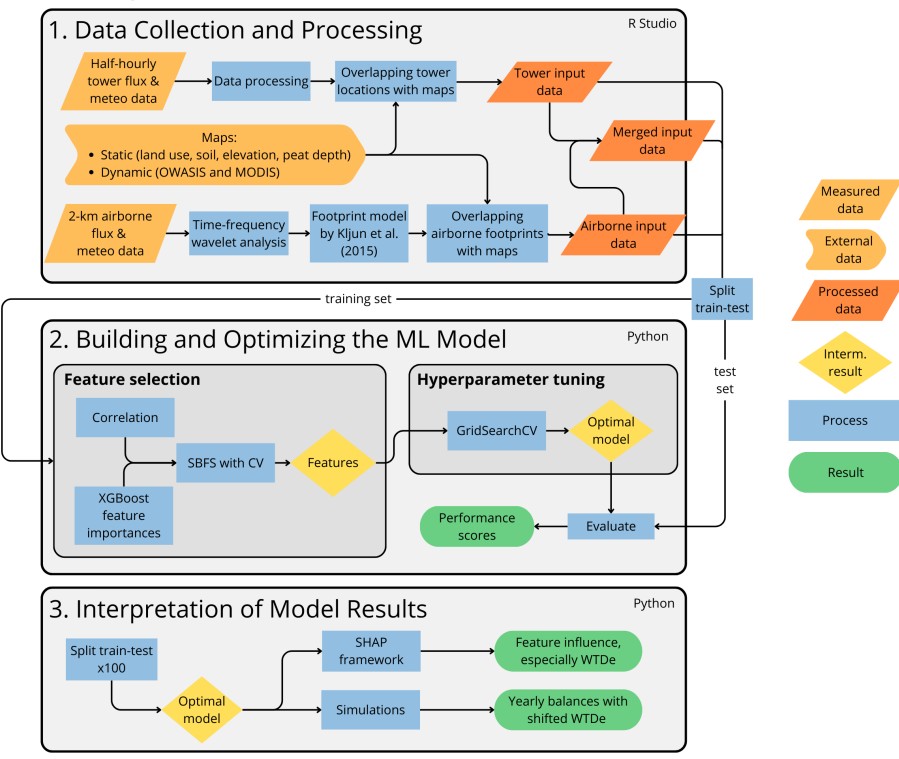

**Figure 1.** Methodological steps for the current study. We divide the methods in three main parts: (1) Data Collection and Processing; (2) Building and Optimizing the Machine Learning (ML) Model; (3) Interpretation of Model Results. We carried out all steps for the three study areas separately, i.e. for the Groene Hart, southern Friesland, and the western part of Overijssel (see Fig. 2), as well as for all areas combined. Sources for external data can be found in Table 2; more details on data processing can be found in 2.1.3 and Appendix A. All steps are described in the text.

gas emissions from drained fen meadows, their drivers, and the efficiency of proposed mitigation measures. Here, we test a machine learning approach on the combination of airborne and tower data to assess the most important drivers of $CO_2$ emissions on a regional level, and to quantify the influence of water table depth.

In Fig. 1, a visual overview of the methodology is shown, which can be roughly split up into three parts: data collection and data processing; building and optimizing several machine learning models; and interpreting the results with Shapley values and simulations. In this section, we describe these three parts of our approach.

## 2.1 Data Collection and Processing

### 2.1.1 Study area

The study area comprises the three main peat soil areas in the Netherlands: the Groene Hart in the south west of the country, southwest Friesland, and the 'Kop van Overijssel' south of that (see Fig. 2). These peat areas have entirely been formed during the Holocene, reaching their maximum extent (about 50% of present Netherlands) around 4000 years ago. Between 2000 and 1000 years ago, large tracts eroded away by a rising and repeatedly intruding sea. Since medieval times, peat has been extracted by humans, and the land has been drained for agricultural purposes. Peat mining continued at a large scale until the late 19[th] century, while drainage continues to this day (Erkens et al., 2016; van Asselen et al., 2020). Most remaining peatlands are fens, and the very few that can be characterised as bogs are not subject of this study. The fen meadows are primarily used as pastures for dairy farming and currently cover around 7% (ca 290.000 ha) of the Dutch land surface (Arets et al., 2021). Water table depth in the study area ranges from surface level to 150 cm below surface level, with most deeply drained soils in Friesland. The climate is temperate and humid, and the Dutch Meteorological Institution states mean annual temperatures between 9.5 and 11.5°C and annual precipitation between 670 and 1100 mm (KNMI, 2024).

### 2.1.2 Airborne Flux Measurements

The aircraft used is a SkyArrow 650 TCNS, a light weight environmental research aircraft with a push propeller. Weather permitting, i.e. with good visibility and no rain, surveys were done twice a week between March 2020 and December 2023, alternating between the three areas described above. However, between July 2020 and February 2021, and between November 2022 and June 2023, the aircraft did not fly due to technical issues. In total, we used data from 141 flights. Parallel flight-tracks of 2-3 km apart were designed perpendicular to the prevailing wind direction and landscape gradients, to get complete spatial coverage of the area of interest. Figure 3 shows a typical flight trajectory. Mean flying altitude was 60 meters, so built-up areas had to be avoided. The flight transects covered all major soil and land use classes, although the footprints were mostly dominated by pastures on peat soils - as are the areas.

The aircraft was equipped with an open-path gas analyzer for $CO_2$ and latent heat fluxes, a thermocouple for sensible heat, both depending on a BAT-probe for 3D wind speed and momentum flux. In addition, net radiation, incoming and reflected photosynthetic active radiation (PAR), air and surface temperature were measured. Most instruments sampled at either 50 or 20Hz. More detail on the aircraft and its equipment can be found in Vellinga et al. (2013). Post-flight processing started with de-spiking raw data. Next, 50 Hz air pressure and temperature measured by the BAT probe were converted to 3D wind fields and corrected for all aircraft motions. Then, covariances between wind and $CO_2$ concentrations were calculated. Conventionally, covariances are calculated over time and then spatially integrated over a fixed window, for example of 2 km long. However, varying conditions can require different window lengths (Sun et al., 2018). Furthermore, this block averaging method potentially suffers from spectral losses and reduced statistical precision in lower frequencies, as is the case with tower-based measurements using typical averaging times (Paleri et al., 2022). As in Metzger et al. (2013), we used wavelet cross-scalograms calculated over the entire flight to find covariances in the frequency domain: smaller-scale, local turbulent fluxes at high fre-

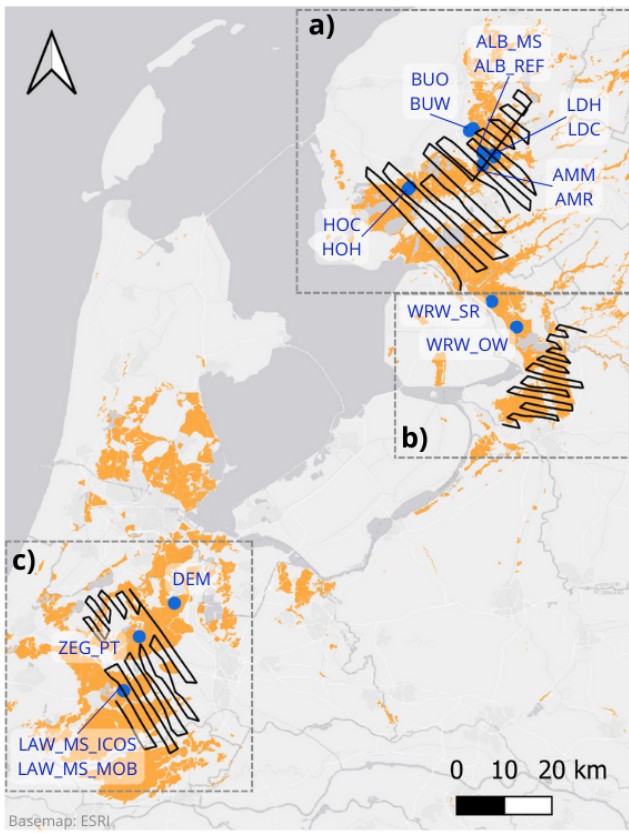

**Figure 2.** The EC tower network used in this study, with three flight tracks over the study areas: (**a**) Friesland, (**b**) Overijssel and (**c**) Groene Hart. Peat distribution is shown in orange. Airborne fluxes are calculated over the entire flight tracks, including the turn, since the banking angle was kept less than 15 degrees. General information on the EC sites and processing can be found in 2.1.3 and Appendix A. For more detailed, site-specific information, see Kruijt et al. (2023). The map was made in QGIS using an ESRI base map; peat distribution was obtained from the soil map (Wageningen Environmental Research, 2024).

quencies, and larger-scale mesoscale contributions at low frequencies with large wavelengths. Wavelengths larger than the boundary layer height were discarded. The fluxes at these two scales are then summed in non-overlapping 2 km windows to
130 get the flux over all scales. Further processing including quality checks and u* filtering was done following the framework of Foken et al. (2004). In Appendix A we provide an overview of the applied steps. In addition, the most important meteorological scalar variables were also averaged over a 2 km spatial window.

To determine the spatial origin of the airborne measurements, the flux footprint model by Kljun et al. (2015) was used. This two-dimensional source weight function is a parameterization of the backward Lagrangian model and describes the spatial
extent, position, and distribution of the contributing surface area. The function can be applied on wide-ranging boundary layer conditions, and has been widely used by studies dealing with airborne flux observations (Hannun et al., 2020; Sun et al., 2023). The input parameters include Obukhov length ($L$), standard deviation of lateral wind velocity ($\sigma_v$), measurement height ($z_m$),

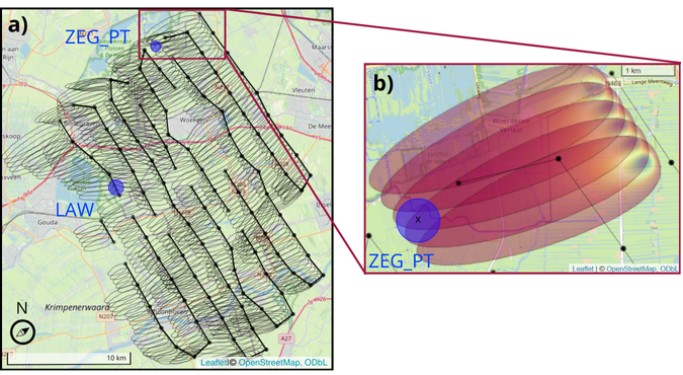

**Figure 3.** Airborne sub-footprints of a typical flight over the Groene Hart, with flight altitude of 60 meters. In this study, we used static circular footprints for towers, which are shown in blue for LAW and ZEG_PT. In (**a**), five sub-footprints are visible for every 2-km window, where the contour lines represent the area from which 80% of the measured flux originates. All sub-footprints were overlaid with spatial data, and subsequently combined and normalized to get the final footprint values. The wind-rose shows the average wind direction. In (**b**), we show the contribution distribution within the first five sub-footprints: blue indicates the highest contribution, red indicates the lowest. The black 'x' denotes the ZEG_PT tower location. In both (**a**) and (**b**), the differences in tower and airborne footprints are visualized.

friction velocity ($u_*$), which were measured directly by sensors on the aircraft, and planetary boundary layer height ($h$), which was extracted from the ERA5 product. Roughness length ($z_0$) is implicitly included in the footprint model by the fraction $u(z_m)/u_*$ (Kljun et al., 2015).

For every 2 km window, five overlapping sub-footprints were calculated and overlaid with various maps, described below. A typical flight with all sub-footprints is shown in Fig. 3. For maps with continuous values, weighted averages were computed, whereas for categorical maps, the fraction of each category in the footprint was calculated. To obtain the final contributions to the flux measurements, the sub-footprints were combined and normalized. Footprints with more than 15% of built up area were excluded (discarding 17% of airborne data), as well as footprints where the dominant land use class is anything else than 'grasslands', 'fens and bogs', or 'summer crops' (discarding 3%). Finally, we had around 10,400 airborne data entries.

### 2.1.3 Eddy Covariance towers

The NOBV implemented an intensive monitoring network of EC towers (see Fig. 2). They are distributed over the three main fen meadow areas mentioned previously and cover a representative range of soil types and water levels. For site descriptions and specifics, as well as the processing of the raw tower data, see the report by Kruijt et al. (2023); below, we use the same site abbreviations.

We used four measurement sites in the Groene Hart (LAW_MS_MOB, LAW_MS_ICOS, DEM, ZEG_PT), ten in Friesland (ALB_RF, ALB_MS, AMM, AMR, BUW, BUO, HOC, HOH, LDC, LDH), and two in Overijssel (WRW_SR, WRW_OW). Although most of these sites are on agriculturally used land, the two sites in Overijssel are in a relatively wet nature area. We used data covering the period from 16-05-2020 to 31-10-2022, although it should be noted that sites vary in data availability.

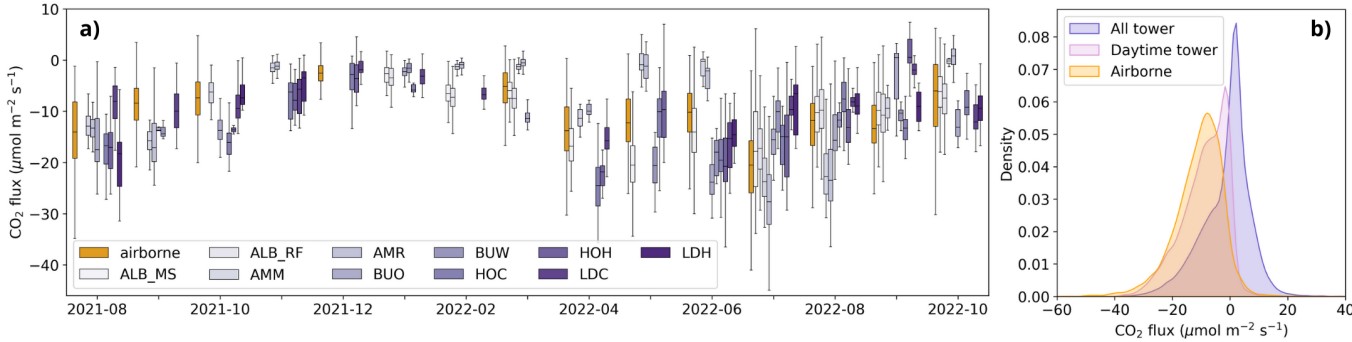

**Figure 4.** (**a**) Airborne and 'daytime' tower measurements between 10:00 and 16:00 of $CO_2$ fluxes in Friesland, binned per month. Boxes represent interquartile ranges, whiskers show minimum and maximum values, excluding outliers. Only a sub-period of the entire dataset is shown. (**b**) Probability distributions of airborne and tower data, for all areas combined. Here, we use the same time frame for 'daytime' as in **a**. The airborne and tower fluxes show similar resemblance in the Groene Hart and Overijssel (not shown here).

At these towers, fluxes of $CO_2$ (and $CH_4$, evaporation and sensible heat, but not considered here) are measured with the EC method, alongside weather station measurements of photosynthetically active radiation (PAR), four component radiation (shortwave and longwave, incoming and reflected), air temperature, relative humidity (RH), rainfall, soil moisture and soil temperature. All sites were equipped with open-path gas analyzers, except LAW_MS_ICOS, which used a closed-path $CO_2$ sensor.

Otherwise, sensor set ups were identical. The equipment was mounted on top of telescopic masts and arranged perpendicular to prevailing southwest winds. Measuring height ranged between 1.5 m and 6 m, based on the desired footprint size. Half-hourly fluxes were calculated using EddyPro (LI-COR Biosciences, 2023) and subsequently post-processed using a series of filters (see Appendix A). All processing was streamlined across towers, and no attempts at gap filling were made. Outliers defined as 0.5% highest and lowest values after filtering for $CO_2$ flux were removed, resulting in 66,400 half hourly data records in total.

In Fig. 4, a part of the measurement period in Friesland is shown, comparing monthly airborne data to monthly daytime tower data across multiple sites. Although the airborne and tower data can never be compared directly due to intrinsic differences in footprint sizes, the seasonal trend and magnitude of the NEE fluxes is similar. Moreover, for most months the aircraft data are within the range observed by the towers, with both towers exhibiting monthly fluxes lower than aircraft and towers showing monthly fluxes larger than observed by the aircraft. This suggests already that the aircraft measures a truly mixed signal.

Similarly to the airborne data, the tower data was overlaid with various maps, but without using the footprint model by Kljun et al. (2015). We expected the differences between individual 30 min tower footprints to be negligible compared to the airborne footprints, so we set a fixed circular footprint for every tower (see Fig. 3). The radius was based on the average 80% of the footprint distance, which was given in the dataset. This footprint was used to extract corresponding spatial data with high spatial resolution (<200 m). Because of relatively consistent study sites (i.e. flat pastures) and relatively low towers (up to 6

meters), the areas within these footprints were largely homogeneous, with the exception of ditches, which were not accounted for. For maps with spatial resolution as low as 250 m, the average value within a radius of 500 m from the measurement point

**Table 1.** Number of datapoints per datatype and area, rounded to the nearest hundred. The 'merged' datasets consists of the airborne and tower datasets.

| Datatype | Groene Hart | Friesland | Overijssel | All |
|----------|-------------|-----------|------------|--------|
| Tower    | 21 100      | 30 300    | 15 000     | 66 400 |
| Airborne | 4 100       | 3 700     | 2 600      | 10 400 |
| Merged   | 25 200      | 34 000    | 17 600     | 86 800 |

was taken (including >12 grid cells), because the value from a single grid cell could erroneously deviate from surrounding grid cells. Table 1 shows the final amount of data.

### 2.1.4 Supplementary Surface Data

Additional surface data potentially explaining $CO_2$ flux variations was gathered from various sources, including both static and dynamic maps (see Table 2). Static maps generally had a high spatial resolution (5-25m) and included land use, soil, peat depth, and elevation. Water-related information was retrieved from the operational product OWASIS by Hydrologic (2019), which consisted of three daily maps: ground water level with respect to sea level (m BSL), soil moisture in the root zone (mm), and air-filled pore space in the unsaturated zone (mm). This product is at national scale with a spatial resolution of 250 m, and

is based on the national hydrological model (LHM, Ligtenberg et al. (2021)). It is made operational by including evaporation and precipitation data and it is the only water-information product covering the entire study area. It performs well in showing trends, but the absolute pixel-based accuracy is up for discussion, thus we consider a wider area than a single pixel. Lastly, two vegetation indices were retrieved by remote sensing from MODIS: the Normalized Difference Vegetation Index (NDVI) and the Enhanced Vegetation Index (EVI). Satellites Aqua and Terra, each with 16-day revisit time, were combined and linearly

interpolated to obtain daily values.

     The categories of the land use and soil maps were reclassified to obtain a smaller but representative number of variables (see Appendix B). Using the collected information, some additional covariates were calculated, such as effective water table depth (WTDe) based on groundwater level and elevation (Eq. 1), the percentage of all peat classes together present in the footprint (AllPeat) as well as for all peat on sand and peat on peat classes (Eq. 2). Combining peat depth with WTDe, the peat exposed

to air ('exposed peat depth') in cm was calculated (Eq. 3).

$$WTD_e = elev. - OWASIS\_GW \tag{1}$$

$$AllPeat = \sum_{i=1}^{n} peatclass_i \tag{2}$$

$$ExpPeatDepth = min(PeatDepth, WTD_e) \tag{3}$$

**Table 2.** Overview of supplementary data sources. For comparison, tower heights ranged between 1.5 and 6 meters, with footprints spanning several hundred meters; airborne flying altitude is 60 meters, with footprints spanning several kilometers (also see Fig. 3).

| Variable | Spatial res. | Temporal res. | Source |
|---|---|---|---|
| Land use* | 5 m | - | Landelijk Grondgebruik Nederland, LGN2020 (Hazeu et al., 2023) |
| Soil* | 5 m | - | Bodem Data, BOFEK2020 (Wageningen Environmental Research, 2024) |
| Elevation (±5 cm, Actueel Hoogtebestand Nederland (2025)) | 25 m | - | Algemeen Hoogtebestand Nederland, AHN3 (Rijkswaterstaat, 2019) |
| Peat depth (± 10-30 cm, Wageningen Environmental Research (2015)) | 100 m | - | Bodem Data (Brouwer et al., 2023) |
| Groundwater level Air-filled pore space Soil moisture | 250 m | Daily | OWASIS product from Hydrologic, personal communication (retrieved January 9, 2024) |
| NDVI (±0.025, Didan (2025)) EVI (± 0.025, Didan (2025)) | 250 m | 8 days | MODIS: MOD13Q1 Terra and Aqua (Didan, 2021) |

*Variables with an asterisk are categorical.

where WTDe is effective water table depth below surface level; elev. is elevation; OWASIS_GW is groundwater level below sea level; peat class represents the fraction of the footprint in peat soil class $i$; ExpPeatDepth is exposed peat depth, or peat exposed to air. Eq. 2 was also used summing only peat on peat classes, and peat on sand classes (see Appendix B2 for soil classes).

## 2.2 Building and Optimizing the Machine Learning Model

For every area, as well as for all areas combined, we built a machine learning model based on the combination of tower and airborne data. We used Boosted Regression Trees (BRT), as they are increasingly used in environmental studies, and are furthermore recommended by studies analyzing airborne flux measurements (Metzger et al., 2013; Serafimovich et al., 2018; Vaughan et al., 2021). The package XGBoost (eXtreme Gradient Boosting) was used, due to the high predictive performance and computing speed (Nielsen, 2016).

For all sets of input data, we made a train-test division. Commonly, this is done in a random manner, but in our case, because of time-dependency of the data and the potential for data-leakage, this is not recommended. For the airborne data, we created the test set by randomly selecting individual flight-legs. For the tower data, we divided the data in blocks of four weeks, using the first three for training and the fourth for testing. We ensured the final data distribution was around 80% train and 20% test. We expected this to be a good trade-off between avoiding data-leakage and keeping enough data for training. Furthermore,

all features in the training set were normalized. The features in the test set were scaled accordingly, based on the statistical properties of the training set.

For each model, the model was tuned in two steps: *a)* reducing the number of explanatory variables: 'feature selection', and *b)* optimizing the model's settings: 'hyperparameter tuning'. Feature selection was done in a hybrid manner, as is frequently done by studies that use XGBoost (Ogunleye and Wang, 2019; Prabha et al., 2021; Sang et al., 2020; Wang and Ni, 2019). First, we analyzed Pearson correlations to roughly select features correlated to $CO_2$, and to reject inter-correlated features. Second, feature importances embedded in XGBoost were computed. These first two steps served as a pre-selection of features for the third, more extensive method: Sequential Backward Floating Selection (SBFS). SBFS includes an extra 'floating' element compared to the more standard and widely used Sequential Backward Selection (SBS), and is known to give good results (Chandrashekar and Sahin, 2014; Rodríguez-Pérez and Bajorath, 2020). SBFS is more time costly than SBS but simultaneously reduces the risk of missing important feature combinations due to early dropping of a specific feature. SBFS was run with 5-fold cross validation, the model used was a XGBoost tree with n_estimators = 1000, learning_rate = 0.05, max_depth = 6, subsample = 1, and the scoring metric for the algorithm was RMSE. As there are multiple features describing water dynamics, several options were run separately, excluding inter-correlated features in the same run. To avoid unequal representation in different folds of the cross validation, all datasets were shuffled beforehand. To evaluate which subset of features is optimal, the $R^2$, MSE, bias and variance of each model proposed by SBFS were computed, and model parsimony was taken into account. The finally selected features are photosynthetically active radiation (PAR), surface temperature (Tsfc), relative humidity (RH), enhanced vegetation index (EVI), effective water table depth (WTDe) and peat depth.

After optimization of the feature subset, the following hyperparameters were tuned: number of trees (n_estimators); maximum depth (tree complexity, max_depth); learning rate (learning_rate); minimum sum of instance weights in a leaf node (min_child_weight); the ratio of columns when constructing each tree (colsample_bytree); and ratio of instances in every boosting iteration (subsample). We performed a grid search with 5-fold cross validation (GridSearchCV from Scikit Learn) on the training set, the parameter grid is shown in Appendix C. The $R^2$ was used as scoring metric. In addition, we set a monotone constraint on the model to find a negative relationship between PAR and $CO_2$ flux, such that increasing PAR leads to more negative fluxes. Finally, the model with optimized parameters was evaluated using the test set. Here, based on performance metrics, model parsimony, and usability, the final models were selected for each area. To compare the models' power and found relationships from different areas with each other, we also tested models between areas, using the same set of features.

## 2.3 Interpretation of Model Results

As we assume the underlying physical processes steering NEE fluxes are the same, we expect that a model trained on one area can be extrapolated to another. However, we expect the Overijssel model to be different, as all tower measurements in this region are from natural areas, unlike in the Groene Hart and Friesland, where sites are located on agricultural land. Although the Overijssel aircraft flux data covers agricultural land, the amount of airborne data points is limited compared to the tower data (see Table 1).

Beyond optimizing the model for the best $CO_2$ predictions, we wanted to understand the model's functionalities and decisions to infer knowledge on the underlying processes. Here, we used two approaches: the explainable AI tool SHapley Additive exPlanations (SHAP) and annual simulations at our site locations. For uncertainty quantification of these interpretations, we applied bootstrapping: we repeated the partitioning in train and test sets one hundred times through a randomized parameter in the splitting algorithm. We created one hundred models based on these training sets, and used the outputs from those models to assess the uncertainty.

The unified SHAP framework was developed to address the difficult interpretation of 'black box' machine learning models (Rodríguez-Pérez and Bajorath, 2020). The method relies on Shapley values, which determine the individual contribution of each feature to the final model outcome considering the collective contribution of all other features. Sequentially, each feature undergoes a process wherein its contribution to the model is negated by assigning a random value to it, thereby resulting in no added predictive power. By comparing model outputs with and without the contribution of a specific feature, the influence of this feature on the model is isolated (Lundberg and Lee, 2017). To consolidate how the model understands the effect of groundwater level, we try to fit regression lines on its SHAP values.

We assembled half-hourly input data (values for PAR, Tsfc, RH, EVI, WTDe and PeatDepth) for each site over the years 2020, 2021 and 2022 from the various data sources (see Table 2). As our sites contained gaps in meteorological data (PAR, Tsfc, RH), we used publicly available hourly data from Dutch Meteorological Institution (KNMI) and linearly interpolated in time to obtain half-hourly values. By using the available data directly, we prevented inconsistencies in the time series that could arise with gap filling. We used KNMI stations "Cabauw" for the Groene Hart area and "Hoogeveen" for the Overijssel and Friesland areas, assuming that this limited spatial variability of meteorological variables is adequate for our sites. For the dynamic maps, we extracted the values at site locations for each day across the three years. Subsequently, using this continuous dataset of features, we let the model predict every half-hour $CO_2$ flux at every site, as well as under hypothetical scenarios where the WTDe was altered by $\pm10$ cm. These predictions were then aggregated to construct annual NEE balances.

## 3 Results

### 3.1 Optimized model settings: features and hyperparameters

In this study, we trained several machine learning models on airborne and ground-based Eddy Covariance data to predict NEE fluxes from peatlands in the Netherlands, aiming to improve our understanding of groundwater-$CO_2$ dynamics. As a first step, we optimized model features and hyperparameters, to achieve models that were both high-performing and parsimonious. Here, we present these model optimization results. As expected, we found strong inter-correlations between meteorological variables (PAR, temperature, relative humidity), as well as between water-related variables (correlation matrices not shown here). Appendix D shows that $CO_2$ flux is most strongly related to PAR, which is also identified by XGBoost feature importance. Furthermore, temperature and EVI score high in all areas, whereas relative humidity scores high in all areas but Overijssel. The importance of the water-related features varies throughout areas. PeatDepth scores high in Friesland and Groene Hart, but lower in Overijssel and when all areas are combined.

**Table 3.** Hyperparameter results and corresponding scores, before (bef.) and after (aft.) hyperparameter tuning. Here, for each model, the same features are used: PAR, Tsfc, RH, EVI, PeatDepth and WTDe.

| Area | colsample by tree | learning rate | max depth | min child weight | number estimators | sub-sample | $R^2$ bef. | $R^2$ aft. | RMSE bef. | RMSE aft. |
|------|------|------|------|------|------|------|------|------|------|------|
| All | 0.9 | 0.005 | 11 | 4 | 3000 | 0.9 | 0.63 | 0.66 | 5.48 | 5.28 |
| Friesland | 0.8 | 0.005 | 11 | 4 | 3000 | 0.9 | 0.63 | 0.67 | 5.62 | 5.32 |
| GrHart | 0.8 | 0.005 | 11 | 1 | 2000 | 0.8 | 0.61 | 0.66 | 5.29 | 5.10 |
| Overijssel | 0.9 | 0.001 | 11 | 6 | 4000 | 0.8 | 0.17 | 0.58 | 6.90 | 5.30 |

For some soil classes such as "pV" (clayey earthy peat soils) and "hV" (thin peaty earthy peat soils >120 cm deep) we find a surprisingly high correlation with $CO_2$ or high XGBoost Feature Importance. The data distribution of these features is different in the airborne and tower dataset: there are multiple towers where 100% of the footprint is (always) hV, whereas the airborne footprints contain a varying range of values for hV, mostly close to 0. Therefore, these features with strongly skewed distributions in the merged dataset were not taken into further consideration for construction of the models.

Based on the correlation matrices and two ranking plots, a pre-selection of features was made to use in the sequential backward floating selection. All models have known important drivers: PAR, RH, Tsfc and EVI. EVI was selected over NDVI because of better scores, and because of its correction for aerosol influence (Huete et al., 2002). Additionally, we included information on water and peat, but since we have various (correlated) features representing this information, we ran SBFS separately for each set of non-correlated features. The sets of features that we added to the drivers named above are: peat depth

combined with one water related feature (water table depth, soil moisture and air-filled pore space), and exposed peat depth, resulting in four separate SBFS runs.

       SBFS showed that as long as the feature set contains information on the daily and seasonal cycles, information about peat, and some water-related feature, the scores are very similar. The models for all areas, GrHart and Friesland all performed well, with little difference in scores. Overijssel's model performed less well. For every model, we selected two to three best

performing feature subsets, and these continued to next step in model optimization: hyperparameter tuning. However, also with optimized model settings, there was a minimal difference in performance with slightly different feature subsets. Hence, we selected six robust features, and tuned hyperparameters for every model with these features, to better enable comparisons between areas. This way, different results cannot be a consequence of different features used. The final features are: PAR, Tsfc, RH, EVI, PeatDepth and WTDe, and the optimized hyperparameters with corresponding scores are shown in Table 3.

Figure 5 shows how the models trained on different areas perform when tested on another area. Although the *Friesland* model is the 'best model' in terms of $R^2$ and bias, we see that the model based on all data performs better overall when applied on the test sets of individual areas. Generally, models trained on specific areas have worse scores when predicting for other regions. Simulating the Overijssel data by models from other areas, and vice versa simulating the other areas with the Overijssel model results in the lowest $R^2$ scores, and the highest biases. For further analyses, we use the model based on all regions.

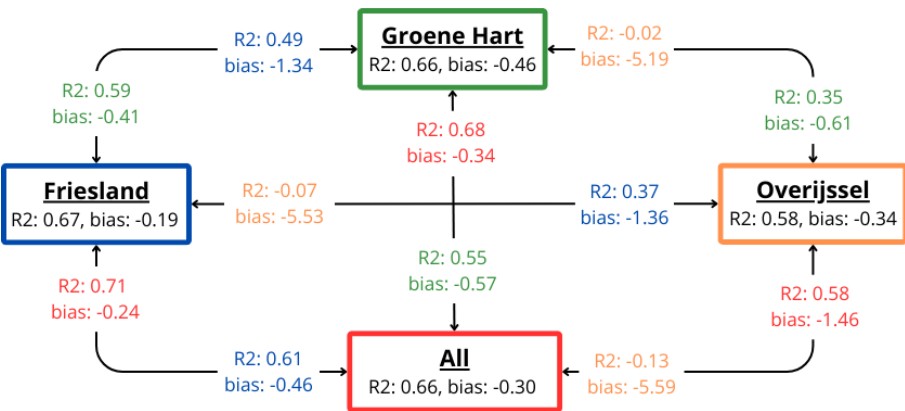

**Figure 5.** Machine learning models trained on data from different areas and corresponding performance scores when tested on the other areas. The same color is used for predictions from the same model. Here, the same features are used: PAR, Tsfc, RH, EVI, PeatDepth and WTDe, and the optimized hyperparameters are shown in Table 3.

## 3.2  Environmental response functions for $CO_2$ flux identified by Shapley analysis

To check the physical consistency of the trained model against prior knowledge and to understand how the model operates we analyze Shapley values for each of the selected features. Figure 6 shows an overview of all Shapley values for the model of all areas. A positive SHAP value indicates that the feature value has a positive contribution to the flux, i.e. increasing emissions or decreasing uptake, and a negative Shapley value vice versa. Increasing PAR and EVI drive more uptake and/or less emissions, whereas increasing temperature and deeper water table depth have the opposite effect. The influence of RH and PeatDepth is not immediately clear from this beeswarm plot.

We delve further into the Shapley values for the features, through scatterplots shown in Fig. 7. Figure 7a shows the SHAP values of PAR and reflects the well-known light response curve, as learned by the model. As PAR increases, the contribution to the predicted flux becomes more negative, especially at higher temperatures. Conversely, with low PAR, SHAP values are positive and higher temperatures cause the flux to increase in the positive direction. Surface temperature drives nocturnal emissions (when PAR is 0), and drives day-time emissions once above about 15 °C. Optimal conditions for uptake are at RH between 40% and 80%, whereas drier conditions, correlating with highest PAR values, drive emissions, as well as wetter conditions that correlate with nighttimes. As WTDe increases (i.e. becomes deeper) so does its SHAP value, but at deeper water levels this seems to level off, or even reverse. Below, we examine this in more detail. EVI values above about 0.55 have a clear stimulating effect of $CO_2$ uptake and the more so with higher PAR. EVI <0.55 does not influence $CO_2$ exchange, especially with PAR=0, indicating no clear effect of vegetation on nighttime emissions. This hints on emissions being driven mostly by heterotrophic respiration. Finally, peat depth shows no apparent relation with $CO_2$ exchange, even though its importance is large enough to be selected into the final model.

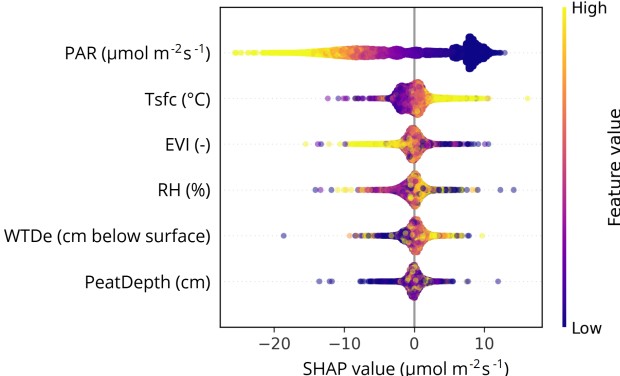

**Figure 6.** SHAP values for all features in the final model, representing the individual contribution of each feature to the final model outcome, depending on the value of that feature. The thickness indicates the amount of data points. For example, there are many data points with low PAR and positive SHAP values, indicating that the model assigns a positive contribution to predicted $CO_2$ flux under low-light (i.e. nighttime) conditions. Abbreviations: PAR (photosynthetically active radiation), Tsfc (surface temperature), EVI (enhanced vegetation index), RH (relative humidity), WTDe (effective water table depth).

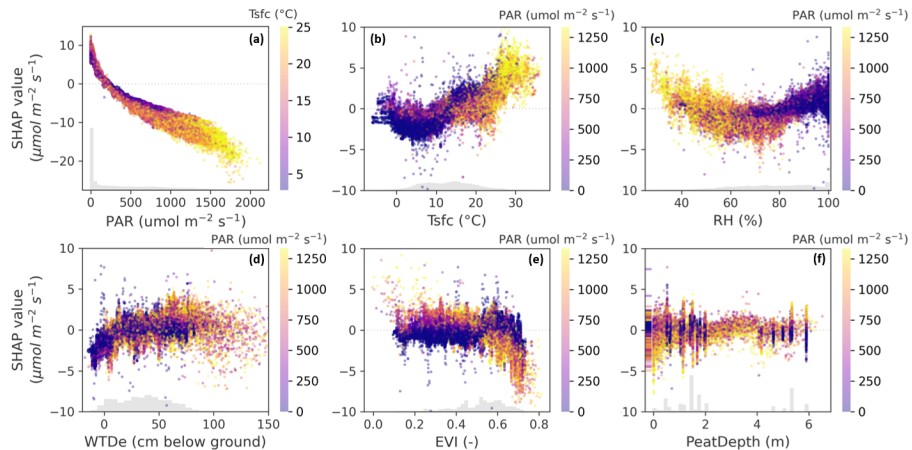

**Figure 7.** SHAP values of (**a**) PAR, (**b**) Tsfc, (**c**) RH, (**d**) WTDe, (**e**) EVI and (**f**) PeatDepth (for abbreviations, see text or Fig. 6). The plots are colored by the values of another feature (Tsfc in (**a**), PAR in (**b-f**)), which in some cases correlates with the depicted one, due to diurnal or seasonal covariance. Nonetheless, the SHAP values represent the effect of only the selected feature on the x-axis. The color gives insight on the conditions in which this effect is happening. The vertical lines in plot ((**f**)) originate from the towers, as a static peat depth map was used. The amount of scatter indicates the robustness of the found relationships.

As we are specifically interested in WTDe, Fig. 8 shows fitted linear, parabolic and Gompertz lines on the bootstrapped
Shapley values for WTDe. We tried fitting several other functions: bell-shaped functions, including and excluding a plateau in the middle; piecewise functions; sigmoid; logistic; shepherd, and more, but the depicted three performed best on our data,

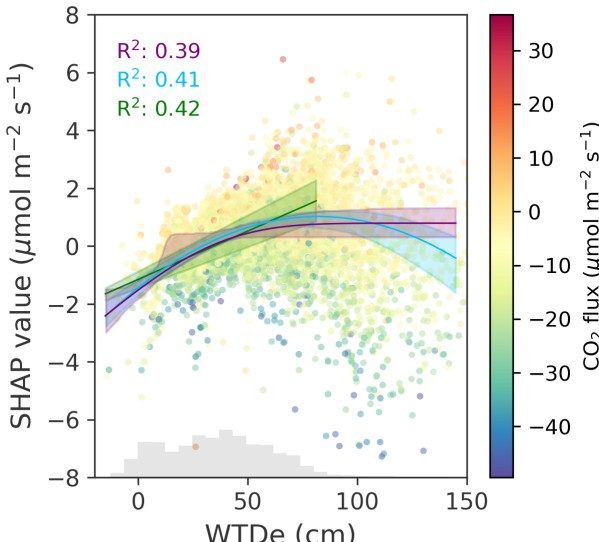

**Figure 8.** SHAP values with fitted linear, parabolic and Gompertz functions, colored respectively green, light blue and purple. Linear regression line stops at the peak of the parabola. The lines are drawn using the medians of 100x fitted parameters on the bootstrapped Shapley values. The shaded areas represent the 90% confidence interval, based on the 5% and 95% of predictions at every WTDe-value. The $R^2$ scores are the means of 100 regression lines. See Table 4 for the fitted parameters.

where WTDe ranges from -0.2 to 1.5 m. The linear regression stops at the peak of the parabola, at WTDe of 0.8 m. Similarly, the Gompertz function seems to truly flatten at this depth. However, the characteristic horizontal part at the beginning of the Gompertz curve is not visible in our data. Despite having different shapes, all three regression lines have approximately the same coefficient of determination (see Table 4). Up to 50 cm, the increase in emissions is coherent.

### 3.3 Simulated response of $CO_2$ fluxes to water table dynamics

For all our sites, we assembled half-hourly input data for 2020-2022. Inspecting the annual course of WTDe at the Overijssel sites, we found there is a systematic underestimation of the water table depth, i.e. OWASIS gives values associated to much drier conditions than the WTDe measurements. As a result, the average annual WTDe for Overijssel was the deepest of all our sites, while these sites are in wet nature areas. Hence, we discarded the simulations for the Overijssel sites. The WTDe at other sites was acceptable, although extremely deep summer water tables were often underestimated.

Letting the model predict the fluxes and calculating the annual balances for the sites in Friesland and the Groene Hart resulted in Fig. 9. For every site-year, there are three dots: one based on the actual WTDe values, one where we subtracted 10 cm, and one where we added 10 cm to every value of WTDe. Therefore, the triplets represent the sensitivity of the site's annual NEE balance to 10 cm change in WTDe. The sensitivity varies from site to site, but all sites combined, there is a curvilinear increase. Fitting a linear regression line yields a slope of 5.3 t $CO_2$ per ha$^{-1}$ yr$^{-1}$ per 10 cm WTDe below the surface. However,

**Table 4.** Fitted parameters for the linear, parabolic and Gompertz curves. The values represent the medians of the 100 fits based on boot-strapped models. The parameters in black correspond to $CO_2$ fluxes in $\mu mol$ $CO_2$ $m^{-2}$ $s^{-1}$, whereas the parameters in grey correspond to fluxes in ton $CO_2$ $ha^{-1}$ $yr^{-1}$. $x$ is WTDe in cm. As the slope in t $CO_2$ $ha^{-1}$ $yr^{-1}$ is one of our primary results, we present its confidence interval here: 4.64 (3.95, 5.39) $* 10^{-1}$ per cm WTDe, while confidence intervals for all fitted parameters can be found in Appendix E.

| Reg. type | $R^2$ | Function | Parameters |
|---|---|---|---|
| lin. reg. up to $\pm$ 80 cm | 0.42 | $y = ax + b$ | a: 3.34 $* 10^{-2}$, b: -1.15 <br> *a: 4.64 $* 10^{-1}$, b: -16.0* |
| parabola | 0.41 | $y = A \cdot (x - x_0)^2 + B \cdot (x - x_0) + C$ | A: -3.65 $* 10^{-4}$, B: 5.62 $* 10^{-2}$, C: -1.14, $x_0$: 5.18 <br> *A: -5.07 $* 10^{-3}$, B: 7.81 $* 10^{-1}$, C: -15.9* |
| Gompertz | 0.39 | $y = S_{max} + S_{diff} \cdot e(-a \cdot e^{(-b \cdot x)})$ | $S_{max}$: 0.79, $S_{diff}$: -10.2, a: 1.48, b: -1.65 $* 10^{-2}$ <br> *$S_{max}$: 11.0, $S_{diff}$: -142* |

*Italic: converted to t $CO_2$ $ha^{-1}$ $yr^{-1}$.*

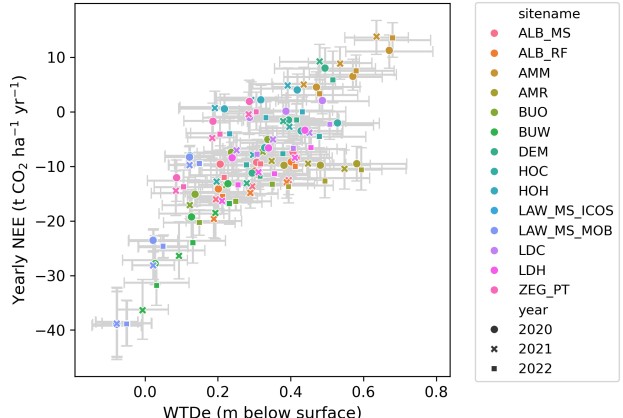

**Figure 9.** Simulated annual $CO_2$ balances for the sites in Friesland and the Groene Hart (see Fig. 2). The annual balances are sums of year-round predicted fluxes at 30 min temporal resolution, using continuous input data from meteorological stations as well as static and dynamic maps. The triplets represent simulations for WTDe - 10 cm, actual WTDe, and WTDe + 10 cm. The uncertainty intervals represent standard deviations, where the vertical intervals are based on 100x annual balances based on the bootstrapped data and corresponding models. Fitting a linear regression line yields a slope of 5.3 t $CO_2$ per $ha^{-1}$ $yr^{-1}$ per 10 cm lowering of WTDe below the surface.

closer examination shows the effect varies per site and per year, and levels off at deeper water levels. We did not extend the simulated effects for deeper water table depths as those scenarios would become unrealistic without altering the other features.

We grouped the sensitivity of predicted NEE fluxes to 10 cm WTDe increase by month, shown in Fig. 10. In gray, the range of sensitivity to 10 cm is depicted that is found in the literature. Although the mean of almost every month falls within the range and does not change sign, there is substantial scatter above and below. Furthermore, there is monthly variation


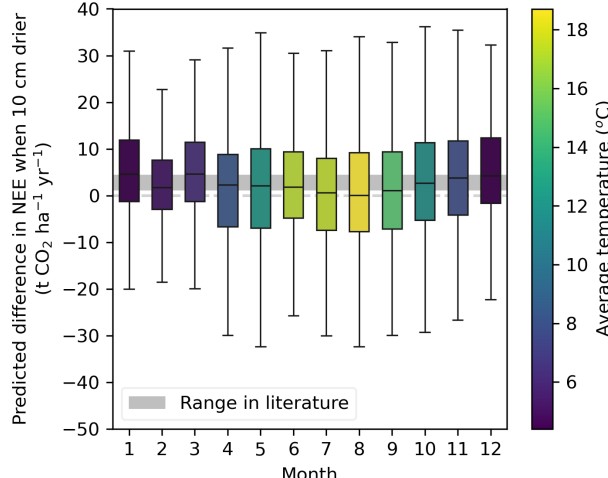

**Figure 10.** Monthly binned differences in NEE predictions when WTDe increased by 10 cm. The color represents the average temperature in our data for that specific month. The 'range in literature' is based on the lowest and highest estimates of the groundwater-$CO_2$ slopes in the literature: 2.1 t $CO_2$ ha$^{-1}$ yr$^{-1}$ per 10 cm as found by Kruijt et al. (2023) and 4.5 t $CO_2$ ha$^{-1}$ yr$^{-1}$ per 10 cm as found by Fritz et al. (2017). See Fig. 11 and Appendix G for other estimates).

in those means, being the lowest in summer, and highest in winter months. We tested whether this variation is statistically significant with a Welch's t-test, which accounts for the unequal variances across groups. Most month-to-month comparisons show significantly different means, but there is no distinct pattern (see Appendix F1). Grouping per season instead of per month

shows that March-April-May (MAM) and June-July-August (JJA) do not differ significantly, but all other season comparisons do, suggesting different effects of 10 cm WTDe increase in autumn, winter, and in spring/summer (see Appendices F2 and F3). If we compare sensitivities to raising WTDe by 10 cm (rewetting), all seasons are significantly different (Appendix F3). Additionally, there is a large variation between sites (see Appendix F4).

## 4    Discussion

### 4.1    Constructed models and their performance

Our final model explains 66% of the observed variance, and is able to provide further insights in key drivers of NEE. The found relationships with PAR, temperature, relative humidity and EVI are in line with physically known processes. The effects of water table depth will be discussed in detail in Sect. 4.3,

     The obtained $R^2$ seems acceptable given the complex interactions analyzed and random noise levels typical for Eddy Covari-

ance observations. Compared to studies also modeling $CO_2$ fluxes but by traditional methods, this $R^2$ is in the same range (Jung et al., 2011; Zulueta et al., 2011; Dou et al., 2018). Similarly, Zhou et al. (2023), who combined satellite data and EC mea-

surements with a Random Forest model found an $R^2$ of 0.6. Still, the $R^2$ is not substantially higher than that of global models (e.g., Jung et al. (2020)), despite the higher information density in our study. We attribute this to the relatively subtle variability within our study area, which encompasses seemingly similar systems in terms of land use, climate, and flux characteristics -

making it more difficult to distinguish patterns than at the global scale. Other studies combining airborne and ground-based data using machine learning approaches tend to have higher $R^2$ (Metzger et al., 2013; Serafimovich et al., 2018; Vaughan et al., 2021), but these all focused on simulating heat fluxes, arguably a simpler process to analyze. Moreover, it appears that none of these studies used separate data subsets for learning and evaluation, as we did in the current study. Evaluating the model on the same dataset it was trained on, would increase the $R^2$ for our best model from 0.66 to 0.89.

Although the Overijssel model performed acceptable, extrapolating to another area gives worse results than taking the mean of fluxes in respective area (negative $R^2$ scores). This does not indicate that the Overijssel area itself is fundamentally distinct. Instead, it is a consequence of the WTDe data not reflecting the wet site conditions well. This might also explain why the *All* model performs better on GrHart and Friesland than when tested on its 'own' test set, which includes Overijssel data, exhibiting the importance of accurate water table data. We trained a model on the combination of only the Friesland and GrHart data,

and it obtained an $R^2$ of 0.67 and bias of -0.26: slightly better results than when the Overijssel data was included. We find that extrapolating between Friesland and GrHart gives reasonable results. Still, we find best results when using the model trained on all data. Overall, we think that these machine learning models, especially the model including GrHart and Friesland data, perform adequately.

The models were well able to find relationships with spatio-temporal variables, especially for key drivers like radiation,

shown by little scatter in Fig. 7a. Additionally, other well-established processes as the influence of temperature and relative humidity are well represented. Furthermore, the SHAP framework enables identification of processes affecting components of NEE that could otherwise not be distinguished. For example, at low PAR, emissions do not increase despite increasing EVI (up to EVI = 0.55), indicating emissions are mostly steered by heterotrophic respiration as opposed to autotrophic respiration. The increase of nighttime emissions for higher EVI values may indicate increased autotrophic respiration. Although additional

research partitioning ecosystem respiration is required, these findings demonstrate the potential of SHAP and ML for process understanding.

While peat depth was important enough to be included in the final model, its SHAP values do not show a distinct effect. This is partly related to the static nature of peat depth and the large amount of data originating from towers, but it also suggests that peat depth, particularly when exceeding the typical range of water table fluctuations, has minimal impact. Correspondingly, it

has been suggested that peat depth exposed to air is a more direct indicator for peat decomposition and thus for $CO_2$ emissions than water table depth (Aben et al., 2024). In our study, we do not find this. The same holds for air-filled pore space, although this property is arguably more complex to model because of peat swelling and shrinking and varying porosity. However, it should be noted that the differences in performance when water table depth is replaced with another water-related feature are minimal, indicating a robust underlying process or relationship that the model is able to find.

We assumed our method of partitioning the data - selecting flight legs and weeks of measurements for the test set and use the rest for training - avoided data leakage. To further examine this, we also applied two other partitioning strategies: one more

stringent, selecting entire flights and sites for the test set, and one less stringent, selecting a random 20% as test set. The former resulted in worse models, but this depended on which sites were left out, as some provide more data and insights than others. Nonetheless, this partitioning strategy should be further examined in future research, because it may better reduce data leakage.

The models based on randomly selected training data all performed slightly better than our the models we use in this study, indicating that our current strategy avoids some data leakage.

In the final optimized hyperparameters, we see that models differ mostly in minimum child weight. Models with a high value for this parameter are more conservative, and end up with fewer splits. We believe the final model based on all areas achieves a good balance between capturing complex patterns and avoiding overfitting. Notably, only the Overijssel model was

strongly improved by hyperparameter tuning. We hypothesize that the low initial performance scores are a result of the smaller dataset size and lower data quality. Increasing the model complexity (by lowering the learning rate and increasing the number of estimators) the model was able to identify relationships nonetheless. However, when attempting to generalize to other areas, it became clear the model was overfitting.

## 4.2   Combining airborne & tower data: pros and cons

Airborne and tower flux data are subject to different errors and uncertainties, intrinsic to their configuration and processing. One of these is flux divergence. Tower data observed only a few meters above a (grassy) surface represent true surface fluxes and any marginal divergence effects are largely corrected through a storage flux correction (Finnigan, 2006). For aircraft fluxes this is less trivial. We aim to minimize divergence errors by flying nominally at 200 ft / 60 m above the surface. This may be considered to be in the 'constant flux layer' given that over all flights the average boundary layer height according to ERA5 re-analysis is

870 m at the time of the flights. Few studies have explicitly addressed $CO_2$ flux divergence: de Arellano et al. (2004) (Cabauw, in the Groene Hart area studied here); Casso-Torralba et al. (2008) (Cabauw again); Vellinga et al. (2010) (Supplementary Material, SW France). Apart from observational constraints to quantify it, $CO_2$ flux divergence is not unidirectional like e.g. sensible heat divergence. The entrainment flux for $CO_2$ can be significant in early morning and may change sign in the course of the day due to $CO_2$ release at night and uptake during the day. Such complications prohibit assessments of $CO_2$ flux divergence

without dedicated observation strategies, let alone allow simple corrections. Neglecting advection, flux divergence equals the scalar storage term, i.e. temporal change in $CO_2$ concentration. From our tower observations we know these to be small around midday. For all these reasons, with Vellinga et al. (2010) and e.g. Meesters et al. (2012) we assume $CO_2$ flux divergence errors are arguably smaller than other errors, not of constant sign (so partially cancel out) and therefore they are neglected here.

As discussed, airborne and tower data have different qualities: airborne data is spatially exhaustive but temporally limited,

and tower data is temporally continuous but with a limited spatial extent. On a given day, the airborne data provides a gradient of seasonally varying landscape features (e.g. WTDe and EVI), as opposed to point values at towers, and represents the entire area, making the input data more diverse. Because of their complementarity, we were able to develop an ML model that includes information on the daily cycle as well as extends beyond the tower locations. Nevertheless, practicalities when combining tower and airborne data may lead to some spurious correlations between features and target, due to specifically two aspects. Firstly,

tower data includes night-time measurements while airborne data is only collected during the day, resulting in differences for

weather-related features (PAR, RH, Tsfc) and almost all positive $CO_2$ fluxes stemming from tower data. Secondly, land use and soil classes are fixed for the tower data and variable for the aircraft data. Together, this led us to omit features from merged models that had too distinct distributions in airborne and tower data, such as the soil class "hV" (thin peaty earthy peat soils of >120 cm deep). Although we prevented artifacts by excluding these features, we also omitted possibly valuable information. We created a categorical variable with the most prevalent land use or soil class in the footprint, but this did not improve model performance. Hence, our results underpin what has been previously found: the relationship with WTDe holds regardless of the land use or soil class (Evans et al., 2021; Tiemeyer et al., 2020).

Another significant difference between the airborne and tower datasets in our study is their size: the former has 10,400 records, the latter 66,400. Consequently, the merged dataset consists mainly of tower measurements, and if we construct a model only based on tower data, we see very similar performance of the two models. In a preliminary, unpublished study we conducted using only the Groene Hart data, the addition of airborne data significantly improved the ML model ($R^2$ increased from 0.47 to 0.61). However, in that case we had restricted datasets available: 7,900 records for the tower data, originating from only two towers and spanning 6 months, and 2,600 records for the airborne data (spanning 18 months). Under similar circumstances, when the tower dataset lacks spatial and temporal coverage, we believe the inclusion of airborne data can improve the model substantially. This was also demonstrated by Metzger et al. (2021), who showed that airborne measurements in combination with pre-field simulation experiments doubled the potential of a surface–atmosphere study. However, in the current study, as the tower network had been seriously extended resulting in much higher spatial coverage, this was not directly visible in an improved $R^2$. We examined the airborne data's added value in three ways.

First, we tested excluding several towers from the training process, to determine if we could replicate the added value of airborne data as in the previous study, and we compared resulting tower and merged models. The outcomes were highly variable, because each tower has different qualities, lengths of measurements, presence of gaps, etc., but removal of certain towers showed similar results as in our previous study. Second, we trained one model with equal amount of airborne and tower instances, which attained an $R^2$ of 0.62. Training the model only airborne data on the other hand, resulted in an $R^2$ of 0.37. This suggests that it is not only the high amount of data that is beneficial for the model, but that there is also intrinsic value in adding tower measurements to airborne. Third, we let all models predict for airborne, tower and merged datasets. The difference between the merged and tower model shows when they are tested on the airborne data: the variance in airborne data was explained for 34% and 10% by the merged and tower models, respectively. The airborne data represents regional fluxes originating from across the entire area. Hence, to model these regional fluxes, extending beyond the locations of the measurement sites, it is necessary to include airborne measurements in the model. Future research could investigate where the trade-off is: when do airborne measurements provide significant complementary benefits to tower data? With *x* number of towers of time *y*, how many flights should be done to obtain enough information? The costs of tower and airborne maintenance should be included. This research could shed light on the most efficient measurement strategies in areas with limited access to resources or with inaccessible terrains.

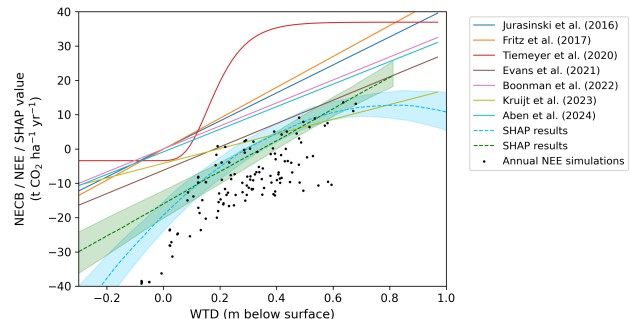

**Figure 11.** Comparison between current findings and literature studies on $CO_2$ flux vs water table depth. Three types of results are shown: regression lines from the literature based on annual NEE or NECB estimates; the dashed lines represent the linear and parabolic fits on the SHAP values; the simulated annual NEE totals are visualized in black, in three-fold per site (WTDe –10 cm, actual WTDe, and WTDe + 10 cm). The shaded areas represent the 90% confidence intervals of the SHAP regressions, based on the 5% and 95% of predictions at every WTDe-value using the bootstrapped models. SHAP regressions are based on direct WTDe values, whereas literature studies and site simulations use annual averages. All fitted regressions can be found in Appendix G.

## 4.3 Influence of water table depth

### 4.3.1 Found relationship compared to previous estimates

Here, we compare our findings to groundwater–$CO_2$ emission relations found by previous studies, shown in Fig. 11 (parameters can be found in Appendix G). Note that although the literature relations, SHAP results and site simulations share the same units (t $CO_2$ ha$^{-1}$ yr$^{-1}$), they represent different types of results. Regression lines of Boonman et al. (2022); Aben et al. (2024); Kruijt et al. (2023); Evans et al. (2021) are based on NECB estimates, and thus include corrections for annual carbon import and export through manure and harvest. In addition, they investigate different locations, partly use different measurement methods - chamber or tower - and they are based on multi-site comparisons thus indicate mostly spatial dependencies. The regressions based on SHAP values do not distinguish between temporal or spatial influence, and depict the effect of changing WTDe on NEE flux as understood by the model, rather than actual $CO_2$ budgets. The annual NEE estimates on the other hand represent the sum of year-round simulated $CO_2$ fluxes with varying WTDe, neglecting carbon import and export as they are not part of the ML model.

Both our NEE estimates and SHAP regression lines show more negative values than those reported by other studies. Firstly, for the annual NEE estimates, this can be partially explained by the difference between NEE and NECB budgets. On highly productive grasslands the export of harvest can be significant, as for example shown by Kruijt et al. (2023) where both NEE and NECB are reported for the same pastures as in the current study. Generally, the carbon in harvested biomass is released back to the atmosphere within a year, mostly in close-by areas. However, these emissions are not measured by the EC towers, though the aircraft might measure them when flying over e.g. barns. Furthermore, simulated fluxes are underestimated due to the negative model bias, which mounts up to 4.2 t $CO_2$ ha$^{-1}$ annually. This bias may partly result from the lack of data

on mowing events: the model continues to predict uptake as usual after mowing, whereas in reality, uptake stops immediately after grass removal (see Appendix H). Secondly, for SHAP values, the reference population mostly determines where SHAP=0, hence the change in impact on the model (i.e. the slope) is more meaningful than the exact level of impact (i.e. the intercept).

Apart from the negative offset, we find very similar relations to current estimates: per 10 cm WTDe, emissions increase annually with 4.6 t $CO_2$ ha$^{-1}$ based on SHAP and 5.3 t $CO_2$ ha$^{-1}$ based on annual NEE estimates. Since the annual NEE estimates are primarily based on farms with distinctive water table management - likely not representative of the entire area - we argue that the SHAP slope, which is based on the full dataset, is a better estimate. Considering the different approaches used, clarifying the negative offset, the correspondence with the literature is remarkable. It suggests that the underlying WTD-$CO_2$ relation persists, also when harvest and manure are disregarded. This may be because we investigate the regional scale, where local fluctuations balance each other out. Another explanation might be that harvesting biomass has mostly short-term impacts on the system (see e.g. Appendix H). By the combination of aircraft measurements, instantaneous data rather than annual totals, and machine learning, we were able to extract this fundamental groundwater–$CO_2$ emissions relationship.

### 4.3.2 Non-linearity of the found relationship

We did not find a linear relationship over the entire range of WTDe in our data (see Figs. 8, 9 and 11). Emissions undoubtedly rise with deeper WTDe, but deeper than 0.8 m, they cease to increase. Based on the SHAP explanations, the optimum-based curve explains the data slightly better than the Gompertz curve, indicating a decrease rather than a stabilization of the effect at deeper water levels. We lack sufficient data at these deeper water levels to make a concluding statement, but given that our datapoints represent instantaneous NEE fluxes instead of annual estimates, it is entirely plausible that a curve with an optimum indeed better represents the underlying process. For example, in conditions with WTDe >0.8 m, moisture conditions can be sub-optimal for peat decomposition. Nijman et al. (2024) and Campbell et al. (2021) found similar response curves comparing nighttime ecosystem respiration to respectively water-filled pore space and volumetric soil moisture content. In these studies, as well as in the current study, instantaneous measurements are compared, as opposed to the studies discussed above. Furthermore, the relationship between WTDe and $CO_2$ flux with deeper water levels is less direct, since the actual soil moisture in the unsaturated zone can vary substantially, which potentially explains the larger scatter in Fig. 8 - in addition to less available training data.

As opposed to Tiemeyer et al. (2020), who finds saturation around a WTDe of 0.4 m, we find the increase to persist until 0.8 m, as was also found by Boonman et al. (2022) and Aben et al. (2024) for average summer WTDe. In the study by Tiemeyer et al. (2020), the amount of data for water levels deeper than 0.4 m is limited. Because we include fluxes on a short time scale and we cover a wider range of WTDe values by using the aircraft, we have a larger dataset to support the observation of increasing emissions up to 0.8 m. Additionally, we do not find an initial plateau at shallow water levels, contrary to what is currently assumed. Although we included wet sites, such as those in Overijssel, the water information product we used was not able to capture these wet conditions and gave incorrect values. Hence, our model was not trained properly on wet sites, and we cannot substantiate the lack of a plateau.

### 4.3.3 Intra-annual and spatial variability

Based on the above discussion, our results are mostly in line with previous studies, with two notable exceptions: the optimum-based shape instead of the linear or Gompertz functions, and the increase of emissions until 0.8 m instead of 0.4 m, also found by Boonman et al. (2022); Aben et al. (2024). A third insight is the intra-annual variation of sensitivity to WTDe change, as shown in Fig. 10 and in Appendix F. Further drainage in summer might not impact emissions as much, whereas drainage in winter has a bigger impact. This might be a reflectance of the monthly varying WTDe: in summer, with very deep groundwater levels, the available oxygen for decomposition is not linearly depending on the groundwater and also determined by the soil structure and its capillary action. As a result, at very deep water levels, the soil in the unsaturated zone can be relatively wet and decrease the effect of further drainage. On the other hand, with more shallow water levels in winter, the presence of roots in the unsaturated zone leads to soil desiccation, promoting oxygen availability down to the water table. Hence, when the water table is lowered, this has a more direct effect on $CO_2$ emissions. Furthermore, we find that raising the WTDe by 10 cm has a significantly different effect across all seasons. Our methodology enables future research on intra-annual occurrences such as season-based mitigation measures and extreme weather events.

In the current study, we applied the simulations on the site locations, which show high variability in responses. These site simulations were partially motivated by the goal of comparing predictions to measurements, as in Appendix H. However, since our model is trained on data covering the entire region — enabled by the use of aircraft data — the simulations can be applied to any location in the area, which can be of interest to policy makers. Although our input data is too coarse to predict the emissions at farm-level, the model can provide insights at municipal or provincial scale in carbon flux dynamics over the years.

### 4.4 Implications for mitigation strategies

Our findings suggest that to reduce $CO_2$ emissions, the optimal water management would be to set the water level as high as possible. To reduce greenhouse gas emissions in general and mitigate climate change, the trade-off with methane should be taken into account (Buzacott et al., 2024). The slope we found lies at the upper limit of current estimates, and our results show that emissions continue to increase down to a WTDe of 0.8 m below the surface, as opposed to 0.4 m in Tiemeyer et al. (2020). Together, these results suggest higher emissions than those that would otherwise be calculated based on previous studies. This would entail that effective mitigation strategies are even more crucial, as the potential for carbon emissions from drained soils may have been underestimated. As such, mitigation measures should not only take into account average annual water table depth, but also the different system behavior throughout the year. Measures should focus on rewetting during the summer and specific attention should go to not lowering the water table during winter. However, the evaluation of potential mitigation measures did not fall within the scope of our study. As we did not incorporate data on mitigation efforts into the model, we cannot draw conclusions in that regard. Nevertheless, this study is part of the Netherlands Research Programme on Greenhouse Gas Dynamics in Peatlands and Organic Soils (NOBV), and contributes to the corresponding measuring, monitoring and modeling framework. Within this framework, the efficiency of mitigation measures is widely studied, using

both data-driven methods as well as process-based models such as SOMERS (Erkens et al., 2022) and findings will be reported to policy-makers.

## 4.5 Recommendations

### 4.5.1 Incorporation of additional data

A general remark on the findings in this study is that the water level data is from a company, that develops and maintains the OWASIS information products together with water boards and knowledge institutes. Comparing OWASIS WTDe to measured WTDe at tower locations, we found that values in summer were often underestimated. Potential sources of errors in the OWASIS data are heterogeneous infiltration capacity within pixel cells, or limitations of remote sensing in capturing deeper soil layers. Despite these uncertainties and underestimations, we consider the OWASIS product suitable for the purposes of this study, as it covers the entire study area, and we have a satisfactory amount of data to balance out pixel-errors.

Still, future research should prioritize including a high-quality soil water product of high(er) spatial resolution. Due to the use of aircraft data, the information on water should be spatially distributed and measurements at tower sites do not suffice. The Netherlands has manifold regional hydrological products that can be used for future regional studies (NHI, 2025). In addition, there are numerous remote sensing products that could be used or combined for water-related information, possibly including ground truth data, as for example was done by Koch et al. (2023) for Denmark, who developed a WTD map for Danish peatlands with a spatial resolution of 10 m. Considering the likelihood that farmers might apply mitigation measures such as changing the water level management or adding a clay layer on the peat, corresponding data would be valuable for subsequent studies. Furthermore, a vegetation index with higher spatial and temporal resolution could be incorporated, such as from Sentinel satellites or from the Dutch groenmonitor.nl website, ideally enabling identification of mowing events. In the current study, we did not directly incorporate mowing or harvest events.

### 4.5.2 Methodological advancements

As mentioned previously, future research should investigate the optimal combination of tower and aircraft data. This way, strategies for the modeling of remote or data-sparse locations can be developed. Herein, a stricter train-test algorithm can be applied. For the tower footprints, we suggest taking a wind-direction based average footprint in the future as opposed to a circular footprint. The SHAP framework has proven highly effective in revealing the processes understood by the ML model. By making the reference population more representative, the base value might become more meaningful. However, the SHAP values do not per definition reflect causal relationships. As an extension to the interpretation by SHAP, therefore, we recommend to include more causality-based and/or physics-based components. Here, we discuss some promising approaches, and our first attempts in applying them. However, they are subject of future plans.

To start, information theory (IT) has already been used in studies to examine causalities of $CO_2$ fluxes (Arora et al., 2019; Farahani and Goodwell, 2024; Yuan et al., 2022). It is a mathematical approach to study the amount of information in a dataset or process based on Shannon entropy: a measure of uncertainty in a system, quantifying its unpredictability. In the current

study, Shannon entropy can shed light on the information in tower vs. airborne data. We computed the Shannon entropy of $CO_2$ fluxes in our tower, airborne and merged datasets (tower + airborne), as well as an entropy-based metric based on Farahani and Goodwell (2024) (see Appendix I for the formula and results). The airborne data has the highest Shannon entropy, indicating the highest level of information in the dataset, followed by the merged data. The results on the metric suggest that the merged model captures the variability in the data better compared to the airborne or tower models. However, the differences are relatively small. Additionally, the results show the tower model is overly complex or noisy, whereas the airborne and merged models slightly under-represent the variability in the data, meaning they smooth over some of the finer details or variability in the $CO_2$ fluxes (Farahani and Goodwell, 2024). At the regional scale, we believe the latter is preferable.

A second option is to alter the loss function in a deep learning model, which is better modifiable than in XGBoost. For example, transfer entropy can be minimized through the loss as is done by Yuan et al. (2022), or directly physically inspired functions or models can be implemented, as successfully done by Liu et al. (2024). New, innovative approaches such as double ML based on causality offer great opportunities for further exploring the greenhouse gas dynamics in drained peatlands (Cohrs et al., 2024).

## 5 Conclusions

In this study, we applied Boosted Regression Trees to learn the relationship between $CO_2$ flux and landscape characteristics from drained peatlands in the Netherlands. We investigated data from the three main fen meadow areas in the country (Groene Hart, Friesland and Overijssel) and finally constructed a model based on all areas combined. We merged $CO_2$ flux data from both airborne and tower measurements, and, to our knowledge, this study is the first to use this combination of $CO_2$ data as input for a machine learning model. The models were optimized with feature selection and hyperparameter tuning, and we accounted for data-leakage by splitting the train and test set based on flight-legs and week numbers. Subsequently, we used the SHAP framework and simulations to assess the influence of most important and relevant environmental drivers.

The method works and the models perform reasonably well with $R^2$ scores between 0.58 and 0.67. We found that extrapolating the model from one region to another performs adequately as long as water table training data is accurate, but that the model including all regions is best for this purpose. As long as the feature subset contained information on the daily and seasonal cycles, information about peat, and some water-related feature, the scores of the models were very similar. Hence, the final, robust features that explained most of the variance in $CO_2$ fluxes are PAR, temperature, relative humidity, EVI, peat depth and effective water table depth (WTDe).

Based on the SHAP values, we find an increase of $CO_2$ emissions until a water table depth of around 0.8 m below the surface. These emissions increase with 4.6 t $CO_2$ ha$^{-1}$ yr$^{-1}$ per 10 cm WTDe on average (90% CI: 4.0-5.4 t $CO_2$ ha$^{-1}$ yr$^{-1}$), which is in agreement with other estimates, albeit at the higher end of the range found in the literature. Together, these results suggest higher emissions related to WTDe than previous studies. Furthermore, we find that an optimum-based function describes the influence of WTDe best within our WTDe range. However, further research using instantaneous measurements on a short time scale (thus including data at deeper water levels) should point out whether the emissions decrease or stabilize after 0.8 m. We

find intra-annual and spatial variation in the response of $CO_2$ flux to 10 cm drying and rewetting. These aspects should be taken into account when developing mitigation measures.

Future research should prioritize including data on water table depth with higher spatial resolution that better captures wet and extremely drained conditions. Causality-based approaches and physics-guided ML models form a promising direction for future studies. A comprehensive comparative study on the synergies between airborne and tower data could contribute to establishing efficient, cost-effective measurement strategies. In conclusion, we have quantified the impact of groundwater changes on $CO_2$ fluxes across drained peatlands in the Netherlands, providing crucial understanding in support of the 1 Mton

reduction target by the Dutch government.

*Code and data availability.* The simulated annual NEE totals and corresponding groundwater levels are available upon request. The input data for the ML models is not yet publicly available due to ongoing research by Bataille et al.. The spatial analysis in R was done using the terra library and the Flux Footprint Prediction (FFP) model by Kljun et al. (2015). In Python, machine learning modeling was done with packages scikit-learn, xgboost and shap. Codes are available upon a reasonable request.

**Appendix A: Processing steps for airborne and tower flux calculation and filtering**

**Table A1.** Processing steps for tower and airborne flux calculation and filtering. EddyPro was used for tower fluxes.

| Processing step | Tower | Airborne |
|---|---|---|
| | Flux calculation | |
| Block-averaging | ✓(30 min) | ✓(2 km) |
| Wavelet decomposition | - | ✓ |
| Reynolds decomposition | ✓ | - |
| WPL[1] correction | ✓ | ✓ |
| Frequency loss correction | ✓ | High freq. not needed because at operating altitude flxues are carried by eddies <10Hz or 4m. Low freq. not needed because of wavelet decomposition. |
| Tilt correction radiation sensors[1] | - | ✓ |
| Flux divergence / storage flux correction | ✓Based on a 1-point profile | - |
| | Filters | |
| Signal strength filtering | ✓Remove Received Signal Strength Indicator (RSSI) < 70 (McDermitt et al., 2011) | - |
| Precipitation | ✓Remove timesteps with precipitation (sensor performance affected) | (no flights with precipitation) |
| Wind direction | ✓Exclude fluxes from undesired wind sectors (site-specific) | - |
| u* filtering | ✓Remove stable night data below u* threshold (site-specific, moving point test by Papale et al. (2006)) | |
| Stationarity and ITC[3] test | ✓ | ✓ |
| Meteorological measurements: physical range filter | ✓ | ✓ |
| Flux magnitude | $-100 < CO_2$ (µmol m$^{-2}$s$^{-1}$) $< 100$ | $-50 < CO_2$ (µmol m$^{-2}$s$^{-1}$) $< 50$ |
| Quality flags | ✓All hard flags by Vickers and Mahrt (1997). ✓Keep only quality flag = 2 (Foken et al., 2004). | ✓Keep quality flags for $CO_2$ and u* < 6 (Vellinga et al., 2013). |

[1] Webb, Pearman,& Leuning (Webb et al., 1980).

[2] Based on aircraft attitude and solar position.

[3] Integral Turbulence Characteristics

## Appendix B:  Reclassification tables

### B1    Reclassification of land use classes

**Table B1.** Reclassification of land use classes.

| New aggregate class | New code | Codes from Dutch national land use map LGN2020 |
|---|---|---|
| Bare soil | bSl | beach (31); drift sands (35) |
| Build up areas | Bld | urban build areas (18); rural build areas (19); forest in build areas (20); forest in rural build areas (22); grass in build areas (23); bare soil in build area (24); main (rail)roads (25); build area (26); grass in rural build area (28) |
| Coniferous forest | cFr | con. forest (12); forest in bogs (40) |
| Deciduous forest | dFr | orchard (9); dec. forest (11); forest in fens (43); tree nursery (61); fruit nursery (62) |
| Fens and bogs | FnB | bogs (20); other marshland (41); reed lands (42); fenmeadow area (45) |
| Grasslands | Grs | nature grasslands (45); grass (1); other land use in rural area (27); salf marsh (30); coastal grass (46); other grass (47) |
| Greenhouses | Ghs | greenhouse horticulture (8) |
| Heath | Hth | open dune vegetation (32); closed dune vegetation (33); dune heath (34); heath (36); sparse grassy heath (37); dense grassy heath (38) |
| Open water | Wat | fresh water bodies (16); salt water bodies (17) |
| Shrubs | Shr | low shrubs in bogs (321); low shrubs in fens (322); low shrubs (323); high shrubs in bogs (331); high shrubs in fens (332); high shrubs (332) |
| Spring crops | SpC | cereals (5); flower bulbs (10) |
| Summer crops | SuC | maize (2); potatoes (3); sugar beets (4); other crops (6) |

### B2    Reclassification of soil classes

**Table B2.** Reclassification of BOFEK2020 soil classes into new aggregate classes for mineral soils.

| New aggregate class | New code | Codes from Dutch national soil map BOFEK2020 |
|---|---|---|
| | | **Mineral soils** |
| Sandy soils | zandG | Zandgronden: 2: Hd30, 16: Zn21, 21: Zn10A, 22: Zn23, 23: Zn40A, 24: Zn50A, 32: Y30, 33: Hd21, 34: Zn50Ab, 35: Zn30, 36: Zn30A, 37: Hn21, 43: pZn23, 45: Hn23, 46: Zn30Ab, 73: cHn21, 84: cHd21, 91: cZd21, 94: cHn23, 96: cY21, 99: cHd23, 107: cHd30, 116: MOb75, 121: MO005, 122: cY23, 134: pZg23, 135: pZg21, 137: Hn30, 140: cHn30, 147: MOb72, 151: cZd30, 157: pZn30, 159: cY30, 171: cZd23, 193: pZg30, 201: pZn21, 204: Zd23, 209: Zb21, 213: Hd23, 218: Sn13A, 224: tZd21, 225: Zb23, 227: Y21, 229: Zd21, 230: Sn14A, 240: Y23, 256: Zb20A, 259: Zd20A, 260: Zb30, 261: Zd30, 262: Zd30A, 263: Y23b, 264: Zd20Ab, 265: Zb30A, 268: tZd23, 272: tZd30, 282: pZg20A, 296: MO002, 300: MOb12, 301: MOb15 |
| Marine clay soils | zeeK | Zeeklei: 3: Mn25A, 27: Mn35A, 30: Mn45A, 39: Mn12A, 48: Mn56C, 54: Mn15A, 55: Mo10A, 56: Mv41C, 57: Mv61C, 59: gMn15C, 60: gMn88C, 61: Mn82A, 79: Mn25C, 80: Mn85C, 85: Mn86C, 87: Mn15C, 89: Mo20A, 106: Mo80A, 110: gMn25C, 113: gMn53C, 125: Mn22A, 139: Mo80C, 148: Mn52C, 154: Mn56A, 156: gMn83C, 160: Mv81A, 161: Mv51A, 166: Mn82C, 167: Mn86A, 170: pMn85A, 172: gMn58C, 173: pMn55A, 176: pMo50, 177: pMo80, 180: pMv81, 194: MK, 210: MA, 222: pMn86C, 228: gMn85C, 236: kMn48C, 241: kMn63C, 251: kMn43C, 271: gMn52C, 274: gMn82C, 275: pMn55C, 276: pMn56C, 277: pMn82A, 278: pMn82C, 279: PMn85C, 280: pMv51, 286: pMn52C, 294: KT, 295: kMn68C, 297: MZz, 298: pMn52A, 299: Mo50C, 302: MZK |
| Alluvial clay soils | rivK | Rivierklei: 4: Rn46A, 5: Rn47C, 6: Rn15A, 8: Rn15C, 9: Rn44C, 10: Rn67C, 11: Rn52A, 12: Rd90C, 13: Rn42C, 15: Rn14C, 17: Rn62C, 18: Rn66A, 19: Rn45A, 20: Rn45C, 49: KRn1, 50: pRn59, 51: Rd10A, 52: KRn2, 81: FG, 83: bRN46C, 128: Rn82A, 129: Rn94C, 130: Rn95A, 136: R040A, 141: Rv01A, 142: Rv01C, 145: KRd7, 162: Rn95C, 165: Ro60C, 178: KRn8, 181: pRn56, 182: pRn86, 192: FK, 195: KX, 208: Rd10C, 220: Rd90A, 266: Ro40C, 267: Ro60A, 270: KRd1, 285: pKRn1, 287: PKRn2, 289: pRn89 |
| Loamy soils | leem | Leem en brikgronden: 68: BZd24, 69: BLb6, 70: BKh25, 71: BKh26, 74: BLd6, 90: BZd23, 92: BLh5, 93: BLh6, 97: BLn5, 98: BLn6, 114: BKd25, 119: BLd5, 120: BKd26, 150: Ld6, 168: pLn5, 186: Lh6, 196: Ldd6, 197: Ld5, 198: Ldh6, 205: Ldh5, 206: Lnd6, 207: Lnh6, 211: Lh5, 212: Ln5, 214: KK, 215: KM, 216: KS, 219: Ln6, 221: Lnd5, 223: Ldd5 |
| Complex associations | gedA | Gedefinieerde Associaties: 58: AP, 66: AZW7A, 67: AZW6A, 75: AO, 76: AS, 78: AZW8A, 95: AQ, 100: AR, 101: AZWOA, 102: AVO, 103: AZW1A, 105: AZ1, 108: AHa, 109: AHc, 111: AHl, 112: AM, 115: AZW5A, 117: AHK, 118: AK, 127: AFK, 155: AFz, 183: ABz, 188: ABK, 231: ABv, 232: ABI, 233: AEm9A, 234: AEm8, 235: AEP7A, 237: AEm9, 238: AGM9C, 242: AAK, 243: AAP, 244: AHs, 245: AHv, 246: AHt, 248: AVK, 249: ALU, 250: AHz, 252: AD, 254: AEk9, 255: AMm, 257: AEm5, 258: AEp6A |

**Table B3.** Reclassification of BOFEK2020 soil classes into new aggregate classes for organic soils.

| New aggregate class | New code | Codes from Dutch national soil map BOFEK2020 |
|---|---|---|
| | | **Organic soils** |
| Sphagnum peat (>120cm deep) | V | Vlierveengronden: 14: Vp, 153: Vc, 158: Vd, 163: Vk, 169: Vr, 202: Vs, 203: Vo, 269: Vb |
| Sphagnum peat on sand (<120cm deep) | Vz | Vlierveengronden, op zand zonder humuspodzol: 253: Vz |
| Sandy earthy peat soils on sand (<120 cm deep) | aVz | Madeveengronden, op zand zonder humuspodzol: 62: aVz |
| Clayey earthy peat soils | pV | Weideveengronden: 42: pVc, 189: pVs, 190: pVz, 281: pVb, 283: pVk, 284: pVr, 288: pVd |
| Thin peaty earthy peat soils (>120cm deep) | hV | Koopveengronden: 47: hVk, 132: hVc, 138: hVd, 146: hVr, 175: hVb, 247: hVs |
| Thin peaty earthy peat soils on sand (<120cm deep) | hVz | Koopveengronden, op zand: 133: hVz |
| Clayey peat soils (>120cm deep) | kV | Waardveengronden: 164: kVc, 184: kVs, 199: kVk, 200: kVr, 217: kVd, 293: kVb |
| Clayey peat soils on sand (<120cm deep) | kVz | Waardveengronden, op zand: 185: kVz |
| Other peat soils | overigV | Overige veengronden: 25: zVc, 26: zVz, 28: zVp, 38: zVs, 64: aVp, 77: aVc, 86: aVs, 144: iVz, 273: hEV, 290: iVp, 291: iVc, 292: iVs |
| Peaty soils | W | Moerige gronden: 1: vWz, 7: ZEZ23, 29: zWp, 31: zEZ21, 40: ZEZ30, 41: zWz, 44: Wg, 53: kWp, 63: bEZ21, 65: bEz23, 72: bEz30, 82: AWg, 88: EK19, 104: AWv, 123: EK79, 124: Wo, 126: EZg23, 131: EZg21, 143: iWz, 149: EZ50A, 152: iWp, 174: EK76, 179: EL5, 187: kWz, 191: EZg30, 226: vWp, 239: uWz |

# Appendix C: Hyperparameter tuning grid

**Table C1.** Parameter grid for hyperparameter tuning.

| Hyperparameter | Values |
| --- | --- |
| n_estimators | 1000, 2000, 3000, 4000, 6000, 7000 |
| max_depth | 5, 6, 8, 9, 11 |
| learning_rate | 0.1, 0.005, 0.001 |
| subsample | 0.8, 0.9, 1 |
| colsample_bytree | 0.8, 0.9, 1 |
| min_child_weight | 1, 2, 4, 6 |

 **Appendix D: Model optimization results**

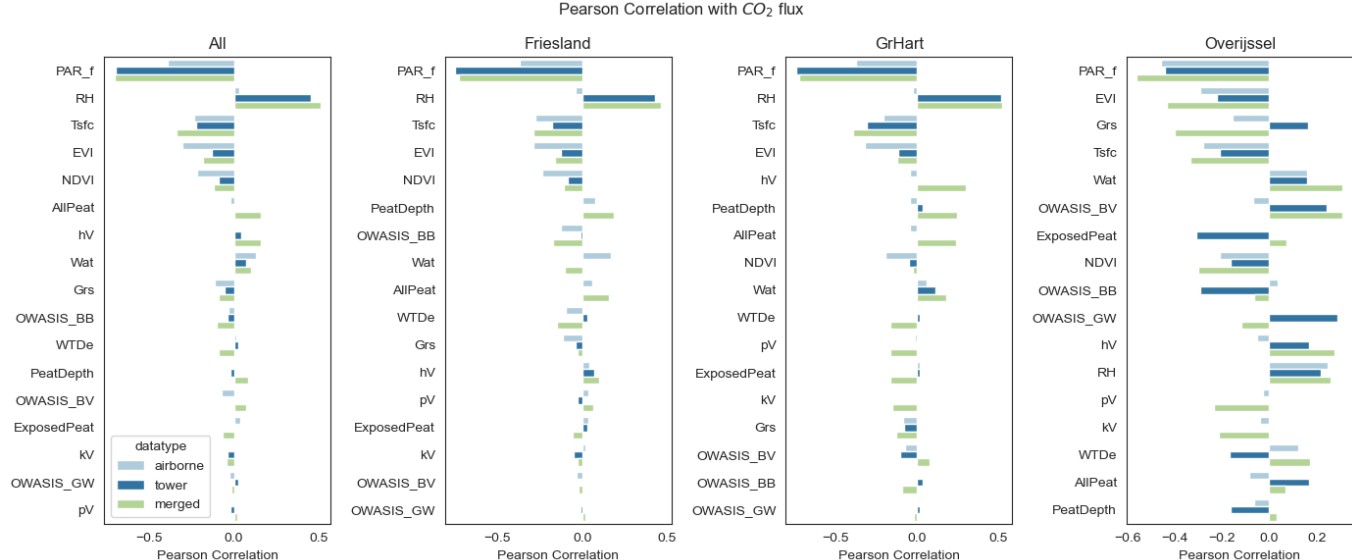

**Figure D1.** Pearson correlation between a selection of features and CO$_2$ flux for airborne, tower and merged datasets per area. The selected features are the most correlated throughout all datasets. The other features (for example, other soil classes and land use classes) are not shown here, for simplicity of the figure. Abbreviations of features: PAR_f: photosynthetic active radiation, RH: relative humidity, Tsfc: surface temperature, EVI: enhanced vegetation index, NDVI: normalized difference vegetation index, AllPeat: percentage of footprint in peat classes, hV: percentage of footprint in soil class thin peaty earthy peat soils (>120cm deep), pV: percentage of footprint in soil class clayey earthy peat soils, kV: soil class clayey peat soils (>120cm deep), Wat: percentage of footprint in land use class open water, Grs: percentage of footprint in land use class grasslands, OWASIS_BB: available open pore space from OWASIS, OWASIS_BV: soil moisture from OWASIS, OWASIS_GW: groundwater level from OWASIS (mbs), WTDe: effective water table depth, ExposedPeat: peat depth exposed to air.

## Appendix E:  Confidence intervals of SHAP regression lines

**Table E1.** Fitted parameters on the SHAP regression lines: LR indicates linear regression, P indicates parabola, G indicates Gompertz. The formulas are shown in Table 4. Values between brackets indicate 90% confidence intervals, based on the 5th and 95th percentile of all bootstrapped regression lines.

| Reg. | Fitted parameters | | | |
|---|---|---|---|---|
| | a | b | | |
| LR | 3.34e-02 (2.84e-02, 3.88e-02) | -1.15e+00 (-1.45e+00, -8.72e-01) | | |
| | *4.64e-01 (3.95e-01, 5.39e-01)* | *-1.60e+01 (-2.02e+01, -1.21e+01)* | | |
| | A | B | C | x0 |
| P | -3.65e-04 (-4.41e-04, -2.81e-04) | 5.63e-02 (1.12e-02, 9.77e-02) | -1.14e+00 (-5.44e+00, 6.88e-01) | 5.18e+00 |
| | *-5.07e-03 (-6.12e-03, -3.91e-03)* | *7.81e-01 (1.55e-01, 1.36e+00)* | *-1.59e+01 (-7.56e+01, 9.56e+00)* | (-4.95e+01, 6.49e+01) |
| | $S_{max}$ | $S_{diff}$ | a | b |
| G | 7.90e-01 (-2.77e+00, 1.29e+00) | -1.02e+01 (-5.62e+01, 4.10e+00) | 1.48e+00 (2.09e-02, 3.21e+00) | -1.65e-02 |
| | *1.10e+01 (-3.85e+01, 1.80e+01)* | *-1.42e+02 (-7.80e+02, 5.70e+01)* | | (-3.62e-01, 3.08e-02) |

*Italic: converted to t $CO_2$ $ha^{-1}$ $yr^{-1}$.*

## Appendix F:  Sensitivity of NEE when WTDe $\pm$ 10 cm

### F1    Welch's t-test on monthly values

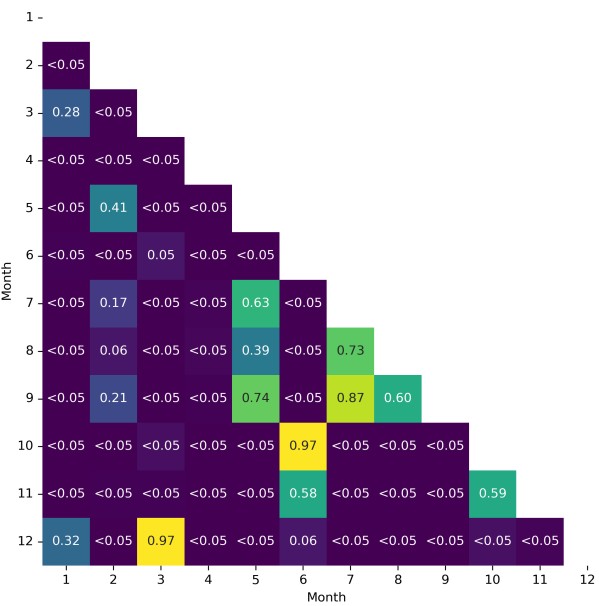

**Figure F1.** Results of p-values for month-to-month comparison of responses to 10 cm drying in WTDe, using Welch's t-test.

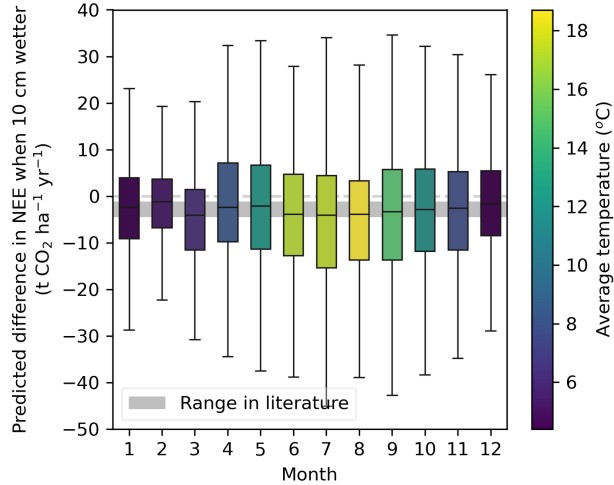

**Figure F2.** Monthly binned differences in NEE predictions when WTDe was raised with 10 cm. The color represents the average temperature in our data for that specific month.

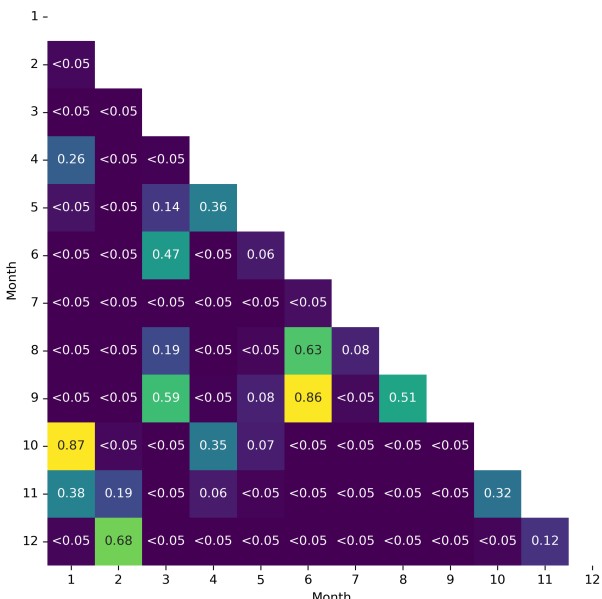

**Figure F3.** Results of p-values for month-to-month comparison of responses to 10 cm rewetting in WTDe, using Welch's t-test.

## F2 Seasonal difference in fluxes after change in WTDe

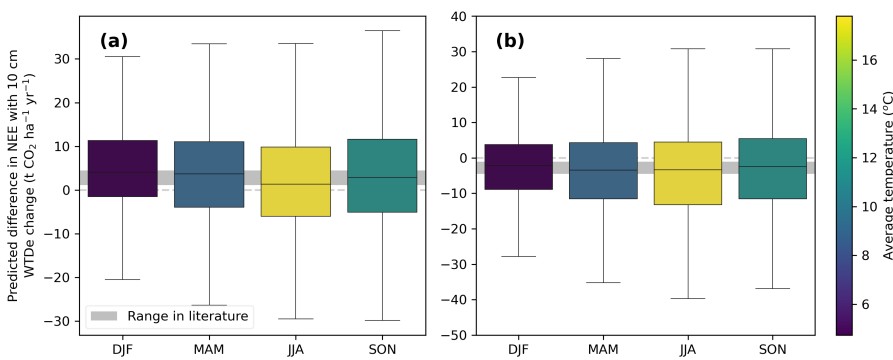

**Figure F4.** Responses of lowering of WTDe by 10 cm of all sites combined, binned per season. Plot (**a**) shows the results for 10 cm drier, (**b**) for 10 cm wetter. Note the different y-axes.

 ## F3 Welch's t-test on seasonal values

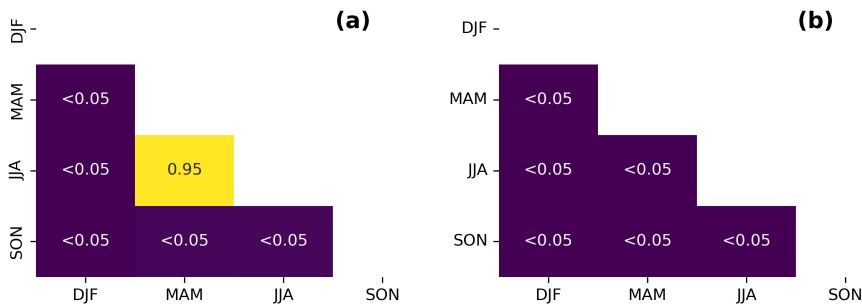

**Figure F5.** Results of p-values for season-to-season comparison of responses to 10 cm change in WTDe, using Welch's t-test. Plot (**a**) shows the results for 10 cm drier, (**b**) for 10 cm wetter.

## F4    Sensitivity per site

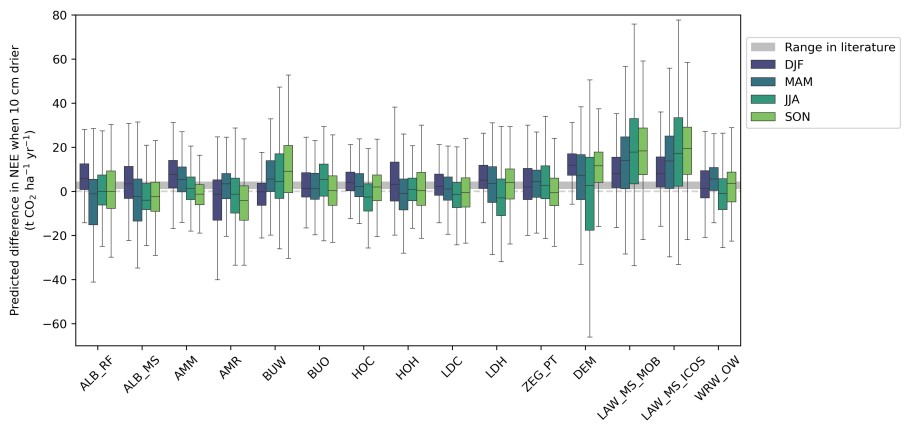

**Figure F6.** Responses of sites to lowering of WTDe by 10 cm, binned per season.

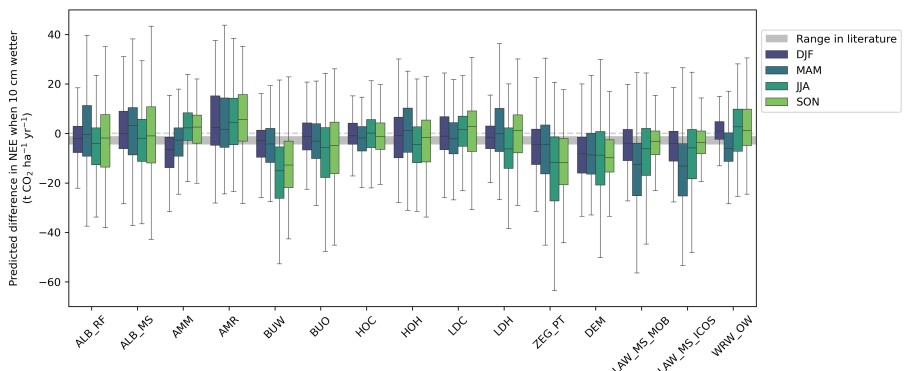

**Figure F7.** Responses of sites to raising of WTDe by 10 cm, binned per season.

## Appendix G:  Relationships found in literature

**Table G1.** Literature studies and corresponding found relationships between emissions in ton $CO_2$ ha$^{-1}$ yr$^{-1}$ and water table depth below surface level in meters.

| Study | Found relationship |
|---|---|
| Boonman et al. (2022) | NECB = 33.5 WTDe |
| Fritz et al. (2017) | NECB = 45.0 WTDe - 0.07 |
| Tiemeyer et al. (2020) | NECB = -3.4 + 40.4 e$^{-7.5e^{-13WTDe}}$ |
| Evans et al. (2021) | NECB = 34.0 WTDe - 6.2 |
| Jurasinski et al. (2016) | NECB = 40.8 WTDe |
| Aben et al. (2024) | NECB = 32.9 WTDe - 0.8 |
| Kruijt et al. (2023) | NECB = 21.4 WTDe - 4.2 |
| SHAP results (current) | NEE = 46.6 WTDe - 16.0 |
| SHAP results (current) | NEE = -51.4(WTDe - 0.05)$^2$ + 77.8(WTDe - 0.05) - 15.8 |
| Simulated WTD sensitivity (current) | NEE = 52.8 WTDe - 25.1 |

## Appendix H: Model predictions and mowing events

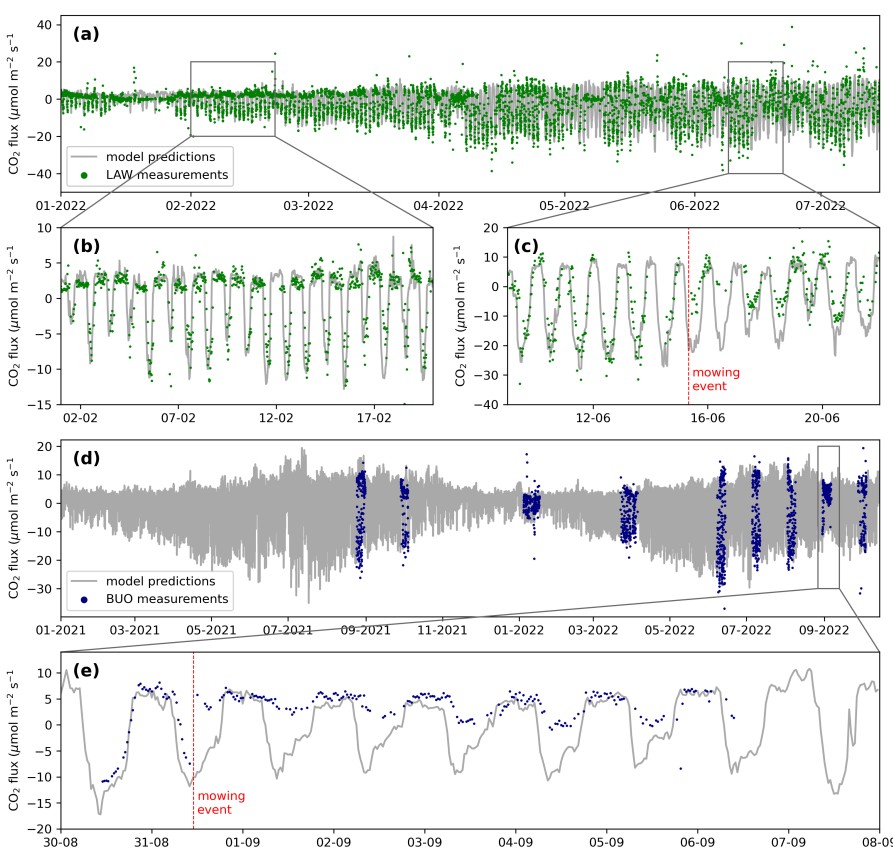

**Figure H1.** Predictions and measurements of $CO_2$ flux at LAW (**a–c**) and BUO (**d, e**). In (**c**) and (**e**), mowing events are highlighted with a red line. In (**e**), uptake immediately stops after mowing, whereas in (**c**), there is still uptake after mowing, which decreases the day after. We hypothesize this is due to the mowing of different parcels at LAW on different days, possibly by the neighboring farmer.

## Appendix I: Shannon Entropy

**Table I1.** Results of Shannon entropy in our datasets for all regions combined.

|  | $H(CO_2flx)$ in training data | $H(CO_2flx)$ in test data (obs) | $H(CO_2flx)$ in modelled data | $A_H$ |
|---|---|---|---|---|
| Airborne | 4.657 | 5.165 | 5.506 | 0.127 |
| Tower | 4.055 | 4.115 | 5.198 | -0.024 |
| Merged | 4.199 | 4.270 | 5.282 | 0.019 |

$$A_H = 1 - \frac{H(CO_{2mod})}{H(CO_{2obs})} \tag{I1}$$

Where H is Shannon Entropy, $CO_2$ mod and obs are modelled and observed $CO_2$ fluxes. The best model perfectly reflects the entropy in the observed data, such that $A_H = 0$.

*Author contributions.* RH, BK and LB guided the scientific process, carried out by LvdP, with respective areas of expertise: regional land-atmosphere interactions and aircraft strategies; ecology, ecophysiology and Eddy Covariance; everything related to machine learning. RH processed the airborne dataset (WCS); LB, BK and JB processed the tower dataset. WF set up the initial footprint analysis script and managed the field-related software. Further processing and analysis was done by LvdP. WJ was responsible for all EC measurement sites, supported by JB. AR carried out a sub-study on Friesland data. RB is responsible for the OWASIS information product. LvdP led the article-writing and AB, YvdV, RH, LB and RB contributed to revisions.

*Competing interests.* The contact author has declared that neither of the authors has any competing interests.

*Acknowledgements.* This study is part of the Netherlands Research Programme on Greenhouse Gas Dynamics in Peatlands and Organic Soils (NOBV). This research program involves Wageningen University (WU), Wageningen Environmental Research (WENR), Vrije Universiteit Amsterdam (VU), Utrecht University (UU), Radboud University, and Deltares Research Institute. We thank all the researchers, technical staff, farmers and landowners involved. We are grateful to Niek Bosma, Reinder Nouta and colleagues of Wetterskip Fryslân for operating mobile sites. The NOBV is funded by the Dutch Ministry of Agriculture, Fisheries, Food Security and Nature and directed by the Foundation for Applied Water Research (STOWA). Additional funds were received from Provincie Fryslân. The Ruisdael Observatory, a scientific infrastructure co-financed by the Dutch Research Council (NWO, grant number 184.034.015) contributed to the aircraft operations. We thank the pilots and staff of Vliegschool Hilversum (VSH) for the careful execution of survey flights, aircraft maintenance, and interactions with the aeronautical authorities.

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
