# Peer review of "Groundwater–CO2 Emissions Relationship in Dutch Peatlands Derived by Machine Learning Using Airborne and Ground-Based Eddy Covariance Data"

_EGUsphere, 2025_

## Author Comment (AC1)

Dear Dr. Metzger,

We want to sincerely thank you for the thoughtful review and constructive feedback you provided. We very much appreciate the time you have taken, and we are pleased that you found the study highly relevant to the biogeosciences community. Your detailed insights and recommendations are very valuable to us - below, we provide point-by-point responses with our answers, reflections and adjustments.

We believe the paper improved thanks to your input.

Laura van der Poel and co-authors

**1. Figure 2**

Please add a scale bar to clarify the geographic extent of the EC tower network and flight tracks (i.e., are we looking at 10s, 100s, or 1,000s of km?).

We added a scale bar and changed the figure to the following:

[Figure]

Figure 2: The EC tower network used in this study, with three flight tracks over the study areas: a) Friesland, b) Overijssel and c) Groene Hart. Peat distribution is shown in orange. Airborne fluxes are calculated over the entire flight tracks, including the turn, since the banking angle was kept less than 15 degrees. General information on the EC sites and processing can be found in 2.1.3 and Appendix A. For more detailed, site-specific information, see Kruijt et al. (2023). The map was made in QGIS using an ESRI base map; peat distribution was obtained from the soil map (Wageningen Environmental Research, 2024).

**2. Figure 3**

The ZEG_PT tower is hard to locate even in the inset. Consider increasing font size or adding an arrow pointing to it.

We adjusted the figure such that readability of ZEG_PT is improved. Furthermore, we cleaned up the figure for clarity and extended the caption:

[Figure]

Figure 3: Airborne sub-footprints of a typical flight over the Groene Hart, with flight altitude of 60 meters. In this study, we used static circular footprints for towers, which are shown in blue for LAW and ZEG_PT. In **a)**, five sub-footprints are visible for every 2-km window, where the contour lines represent the area from which 80% of the measured flux originates. All sub-footprints were overlaid with spatial data, and subsequently combined and normalized to get the final footprint values. The wind-rose shows the average wind direction. In **b)**, we show the contribution distribution within the first five sub-footprints: blue indicates the highest contribution, red indicates the lowest. The black 'x' denotes the ZEG_PT tower location. In both **a)** and **b)**, the differences in tower and airborne footprints are visualized.

**3. Figure 4**

Airborne fluxes seem to show lower amplitude compared to tower fluxes. Have the authors considered vertical flux divergence or other systematic differences in data capture and processing? A cluster analysis could help evaluate whether the two platforms yield comparable flux regimes after controlling for diurnal cycle and differing surface heterogeneity in the footprints. Please also see comment 13 below.

Thanks for raising this issue. Referring to figure 4 we do not concur that airborne fluxes *systematically* seem to show lower amplitude compared to tower fluxes. For most of the 16 months shown here, there are sites with higher as well as lower magnitude fluxes, not only in their median, but also their interquartile ranges do not overlap. Only in 1 out of 16 months (oct21) the airborne fluxes are lower than any of the sites. We have not done a formal cluster analysis.

There is more to be said about flux divergence. First, we aim to minimize this by flying nominally at 200 ft/60m above the surface. This may be considered to be in the 'constant flux layer' given that over all flights the average boundary layer height according to ERA5 re-analysis (for a grid box centered on each of the 3 regions) is 870m at the time of the flights.

Few studies have been done on $CO_2$ flux divergence: de Arellano et al. (2004) (Cabauw, in the Groene Hart area studied here); Casso-Torralba et al. (2008) (Cabauw again); Vellinga et al. (2010) (Supplementary Material, SW France). $CO_2$ flux divergence above typical observation heights of towers is notoriously hard to quantify due to transient effects and possibly missed fluxes from high and low frequency contributions. Further it is complicated by the entrainment flux which for $CO_2$ can be significant and even larger than surface flux in the morning, when night time high-concentration boundary layers are flushed with low concentration air from the free troposphere. In the afternoon the entrainment flux may even reverse sign, as boundary layer concentrations may become lower than free trosoheric values, due to strong uptake (photosynthesis). Thus, also the sign of the flux divergence is not a given (as for the vapor flux). Such complications prohibit assessments of $CO_2$ flux divergence without dedicated

observation strategies, let alone allow simple corrections. Finally, neglecting advection, flux divergence equals the scalar storage term, i.e. temporal change in $CO_2$ concentration. From our tower observations we know these to be small around mid-day. For all these reasons, with Vellinga et al. (2010) and e.g. Meesters et al. (2012) we assume $CO_2$ flux divergence errors are arguably smaller than other errors, not of constant sign (so partially cancel out) and therefore they are neglected here.

We added a few sentences in the discussion of Figure 4 as well as in the discussion.

**4. Table 2**

Please include EC tower measurement heights in the text to facilitate comparisons with remote sensing data resolutions. For example, a 250 m MODIS resolution would match a 250 m tall tower (e.g., Xu et al., 2017) [https://doi.org/10.1016/j.agrformet.2016.07.019], whereas shorter towers may be better represented by higher-resolution products such as Sentinel.

All flux towers are relatively low, with measurement heights ranging from 1.5 to 6 m. The aircraft flies at 60 m height (200ft by special permission of aviation authorities), which is substantially lower than the tall tower by Xu et al. (2017). Given these relatively low observation heights, the higher spatial resolution of Sentinel products may offer improvements for future studies compared to MODIS. We mention potential for Sentinel in Section 4.5.1. and added a few sentences on the towers in Section 2.1.3:

"All sites were equipped with open-path gas analyzers, except LAW_MS_ICOS, which used a closed-path $CO_2$ sensor. Otherwise, sensor set ups were identical. The equipment was mounted on top of telescopic masts and arranged perpendicular to prevailing southwest winds. Measuring height ranged between 1.5 m and 6 m, based on the desired footprint size."

We also specified this in the caption of Table 2: "Overview of supplementary data sources. For comparison, tower heights ranged between 1.5 and 6 meters, with footprints spanning several hundred meters; airborne flying altitude is 60 meters, with footprints spanning several kilometers."

**5. Table 3**

The regional model explains 60% of the variance, which is not dramatically higher than global models (e.g., Jung et al., 2020 [https://www.biogeosciences.net/17/1343/2020/]). Please discuss this apparent incongruity, given the expected higher information density at the regional scale.

Thank you for the comment. We added a few sentences discussing this apparent incongruity in Section 4.1, line 336: "Still, the $R^2$ is not substantially higher than that of global models (e.g., Jung et al. (2020)), despite the higher information density in our study. We attribute this to the relatively subtle variability within our study area, which encompasses seemingly similar systems in terms of land use, climate, and flux characteristics - making it more difficult to distinguish patterns than at the global scale."

**6. Figure 5**

The comparison between regional and local models is insightful but largely confirms expectations. Consider moving the detailed discussion and figures to the appendix and condensing the main text.

We considered your suggestion. However, we believe the figure adds to the main text as it validates our choice of using the model based on all areas. Furthermore, it shows the large difference in the performance of the Overijssel model when predicting for other areas.

**7. Figure 6**

Clarify the sign convention for water table depth (e.g., lower water tables as positive depth vs. negative height). The current color coding and SHAP interpretation may confuse readers without a consistent definition.

We added units for every feature in the figure and extended the figure caption, to the following:

[Figure]

Figure 6: SHAP values for all features in the final model, representing the individual contribution of each feature to the final model outcome, depending on the value of that feature. The thickness indicates the amount of data points. For example, there are many data points with low PAR and a positive SHAP values, indicating that the model assigns a positive contribution to predicted $CO_2$ flux under low-light (i.e. nighttime) conditions. Abbreviations: PAR (photosynthetically active radiation), Tsfc (surface temperature), EVI (enhanced vegetation index), RH (relative humidity), WTDe (effective water table depth).

**8. Figure 7**

This figure helps resolve the confusion in Figure 6 by explicitly stating water table depth in cm below ground. Consider using negative values (e.g., -150 to 0 cm) throughout for better intuitiveness also for non-subject matter experts.

We ensured consistency throughout the text and figures, writing explicitly that WTD is in (centi)meters below the surface. We keep this convention, as this is commonly done by groundwater-$CO_2$ related studies (Aben et al., 2024; Boonman et al., 2022; Evans et al., 2021; Koch et al., 2023).

**9. Figure 10**

This simulation result is the centerpiece of the manuscript. Please revise the phrase "increased with 10 cm" to "increased by 10 cm" for clarity.

Thank you for pointing this out. We changed into "increased by 10 cm".

**10. Figure 11**

The SHAP-derived functional relationships are particularly powerful for integrating direct measurements and non-linear interactions. Consider extending the outlook to include the potential for assessing CH4 fluxes and albedo

effects using the same EC framework, enabling full net radiative forcing (NRF) or CO2e tradeoff evaluations through a consistent methodology.

Thank you for the suggestion. While we think this would be an interesting extension of the study, we doubt whether this is feasible due to the following considerations:

- We do not have airborne fluxes of $CH_4$, only concentrations.
- $CH_4$ fluxes are primarily driven by soil temperature and groundwater level, rather than radiation (Buzacott et al., 2024), resulting in less temporal (daily) variability and making them more difficult to model with a ML approach as used in our study.
- $CH_4$ flux data are subject to stricter quality filtering compared to $CO_2$, which reduces the data available for training.
- Achieving a full NRF balance would additionally require consideration of $N_2O$ emissions, a significant term given the high cattle densities in the Netherlands, which are costly and challenging to measure, especially from the aircraft.

Considering all, we did not add the potential for assessing $CH_4$ fluxes to the outlook.

**11. Line 130**

Clarify whether the "2 km windows" were applied in a moving window fashion, and if so, specify the step size, degree of overlap, and the import of those choices on downstream data integration and results.

The 2 km windows were not applied in a moving window fashion: there was no overlap, as to our opinion the overlap would have compromised the independence of the sequential measurements, with possible unwanted effects in the ML models. We now clarified this on line 130: "The fluxes at these two scales are then summed in non-overlapping 2 km windows to get the flux over all scales."

**12. Line 138**

Vaughan et al. (2021) and Serafimovich et al. (2018) used the Metzger et al. (2012) [https://doi.org/10.5194/amt-5-1699-2012] footprint model, not Kljun et al. (2015). Please correct.

As the sentence on line 138 describes examples of the Kljun et al. (2015) model, we removed Vaughan et al. (2021) and Serafimovich et al. (2018).

**13. Line 178**

Please describe how EC tower data processing (e.g., block averaging, de-spiking, spectral correction, density correction, quality flags) compares with airborne EC data processing (e.g., Wavelet decomposition). This will help readers assess the interoperability of the datasets. Please also see comment 3 above.

We added a table in the Appendix with all processing steps for tower and airborne data.

Table 1: Processing steps for tower and airborne flux calculation and filtering. EddyPro was used for tower fluxes.

| Processing step | Tower | Airborne |
|---|---|---|
| | Flux calculation | |
| Block-averaging | ✔ (30 min) | ✔ (2 km) |
| Wavelet decomposition | - | ✔ |
| Reynolds decomposition | ✔ | - |
| WPL[1] correction | ✔ | ✔ |
| Frequency loss correction | ✔ | High freq. not needed because at operating altitude flxues are carried by eddies $<10$Hz or 4m. Low freq. not needed because of wavelet decomposition. |
| Tilt correction radiation sensors[1] | - | ✔ |
| | Filters | |
| Signal strength filtering | ✔ Remove Received Signal Strength Indicator (RSSI) $< 70$ (McDermitt et al., 2011) | - |
| Precipitation | ✔ Remove timesteps with precipitation (sensor performance affected) | (no flights with precipitation) |
| Wind direction | ✔ Exclude fluxes from undesired wind sectors (site-specific) | - |
| u* filtering | ✔ Remove stable night data below u* threshold (site-specific, moving point test by Papale et al. (2006)) | |
| Stationarity and ITC[3] test | ✔ | ✔ |
| Meteorological measurements: physical range filter | ✔ | ✔ |
| Flux magnitude | $-100 < CO_2$ ($\mu$mol m$^{-2}$s$^{-1}$) $< 100$ | $-50 < CO_2$ ($\mu$mol m$^{-2}$s$^{-1}$) $< 50$ |
| Quality flags | ✔ All hard flags by Vickers and Mahrt (1997). ✔ Keep only quality flag $= 2$ (Foken et al., 2004). | ✔ Keep quality flags for $CO_2$ and u* $< 6$ (Vellinga et al., 2013). |

[1] Webb, Pearman,& Leuning (Webb et al., 1980).
[2] Based on aircraft attitude and solar position.
[3] Integral Turbulence Characteristics

**14. Line 365 & 528**

To my knowledge Metzger et al. (2022) [https://doi.org/10.5194/amt-14-6929-2021] were the first to use combined airborne and tower EC heat and water flux data in ML applications to optimize flight track placement. Clarify how the current work extends this to in-situ CO2 fluxes.

Noted. We changed on line 385: "Under similar circumstances, when the tower dataset lacks spatial and temporal coverage, we believe the inclusion of airborne data can improve the model substantially. This was also demonstrated by Metzger et al. (2021), who showed that airborne measurements in combination with pre-field simulation experiments doubled the potential of a surface–atmosphere study. However, in the current study, as the tower network had been seriously extended resulting in much higher spatial coverage, this was not directly visible in an improved $R^2$. We examined the airborne data's added value in three ways."

On line 528, we specified that we are the first (to our knowledge) to combine these data sources for $CO_2$ fluxes: "We merged $CO_2$ flux data from both airborne and tower measurements, and, to our knowledge, this study is the first to use this combination of $CO_2$ data as input for a machine learning model." We also added a reference to Metzger et al. (2013) in the introduction as a study where airborne and ground-based measurements are combined in a ML framework.

---

## Author Comment (AC2)

Dear Dr. Wiekenkamp,

Thank you for taking the time for this extensive, constructive and insightful review. We highly appreciate both the general feedback and detailed comments, and are very pleased you think the manuscript has high scientific quality. Below, we show the adjustments we made based on your suggestions and comments.

We believe the paper improved thanks to your input. Thank you again for your helpful review,

On behalf of all co-authors, Laura van der Poel

**General Comments**

**Captions**

I find that some of the figure captions do not cover all relevant information (for example Figure 6) and sometimes miss info on the items that are described and/or on the abbreviations (see also Biogesciences info on figure captions – "The abbreviations used in the figure must be defined, unless they are common abbreviations or have already been defined in the text"). However, I do want to stress that other captions in the manuscript provide very good and detailed information about the content of the figure (for example Figure 11). I would suggest to make sure that all figures have captions with similar content and quality.

We ensured clear mentioning of abbreviations throughout the text and extended the following figure captions:

- Figure 1: Methodological steps for the current study. We divide the methods in three main parts: (1) Data Collection and Processing; (2) Building and Optimizing the Machine Learning (ML) Model; (3) Interpretation of Model Results. We carried out all steps for the three study areas separately, i.e. for the Groene Hart, southern Friesland, and the western part of Overijssel (see Fig. 2), as well as for all areas combined. Sources for external data can be found in Table 2; more details on data processing can be found in 2.1.3 and Appendix A." *(This is a new appndix, see point 11.)* ". All steps are described in the text."

- Figure 2, see point 9.

- Figure 3, see point 12.

- Figure 4, see point 14.

- Figure 6: SHAP values for all features in the final model, representing the individual contribution of each feature to the final model outcome, depending on the value of that feature. The thickness indicates the amount of data points. For example, there are many data points with low PAR and a positive SHAP values, indicating that the model assigns a positive contribution to predicted $CO_2$ flux under low-light (i.e. nighttime) conditions. Abbreviations: PAR (photosynthetically active radiation), Tsfc (surface temperature), EVI (enhanced vegetation index), RH (relative humidity), WTDe (effective water table depth).

- Figure 9: Simulated annual $CO_2$ balances for the sites in Friesland and the Groene Hart (see Fig. 2). The annual balances are sums of year-round predicted fluxes at 30 min temporal resolution, using continuous input data from meteorological stations as well as static and dynamic maps. The triplets represent simulations for WTDe - 10 cm, actual WTDe, and WTDe + 10 cm. The uncertainty intervals represent standard deviations, where the vertical intervals are based on 100x annual balances based on the bootstrapped data and corresponding models. Fitting a linear regression line yields a slope of 5.3 t $CO_2$ per ha$^{-1}$ yr$^{-1}$ per 10 cm increase in WTDe below the surface.

- Figure 10, see point 30.

**Title of Results Sections**

The titles of the results sections are more focused on the results of a particular method and are in general pretty generic. In order to guide the readers to a particular section of the results, I suggest using section titles that better reflect the content of each section. For example, instead of using the title "Simulations" I suggest using something like "CO2 Flux Simulations" or "CO2 Flux Simulations and Groundwater Dynamics". Mind that these are just suggestions to illustrate the change in content of such headers.

Following your excellent suggestion, we changed "Simulations" to "$CO_2$ flux simulations and groundwater dynamics. In addition, we changed "Optimized model settings" to "Optimized model settings: features and hyperparameters" and "Shapley Explanations" to "Influence of features on $CO_2$ fluxes through Shapley Explanations".

**Strengthen Key findings**

Even though I find the most important key finding (groundwater effect of fluxes) pretty well positioned in the abstract (and also in the conclusion), sometimes other key findings (for example when talking about the transferability of the models, process understanding obtained from ML SHAP analysis hinting at emission driven by heterotrophic respiration and effects of e.g. PAR– Fig 7, looking at the importance of accurate groundwater information and spatial and temporal variability observed between stations), the main take-aways from the different results sections could be more prominently mentioned in the sections. Sometimes some unexpected findings could also be discussed more in terms of their implications (e.g. why PeatDepth was assigned as a very important aspect, but did not show clear effects on fluxes when looking at the SHAP analysis), but I can also see that quite a lot of discussion is already provided in the current manuscript.

Thank you for the comment. We added the following considerations in the discussion.

In section 4.1, on line 343: "This might also explain why the *All* model performs better on GrHart and Friesland than when tested on its 'own' test set, which includes Overijssel data, exhibiting the importance of accurate water table data."

In the same section we added a paragraph after line 349 describing the SHAP relationships in Fig. 7: "The models were well able to find relationships with spatio-temporal variables, especially for key drivers like radiation, shown by little scatter in Fig. 7**a**. Additionally, other well-established processes as the influence of temperature and relative humidity are well represented. Furthermore, the SHAP framework enables identification of processes affecting components of NEE that could otherwise not be distinguished. For example, at low PAR, emissions do not increase despite increasing EVI (up to EVI = 0.55), indicating emissions are mostly steered by heterotrophic respiration as opposed to autotrophic respiration. The increase of nighttime emissions for higher EVI values may indicate increased autotrophic respiration. Although additional research partitioning ecosystem respiration is required, these findings demonstrate the potential of SHAP and ML for process understanding.

While peat depth was important enough to be included in the final model, its SHAP values do not show a distinct effect. This is partly related to the static nature of peat depth and the large amount of data originating from towers, but it also suggests that peat depth, particularly when exceeding the typical range of water table fluctuations, has minimal impact. Correspondingly, it has been suggested that peat depth exposed to air is a more direct indicator for peat decomposition and thus for $CO_2$ emissions than water table depth (Aben et al., 2024). (...)"

**Code availability**

Information on the code availability is missing in this manuscript. Similar to a data statement, a code availability statement should probably be part of the manuscript. At least to mention what packages are generally available

and what information could be accessed on request.

Following your comment we turned our 'Data availability' statement into a 'Data and code availability' statement and added a few lines regarding the codes in R and Python:

The simulated annual NEE totals and corresponding groundwater levels are available upon request. The input data for the ML models is not yet publicly available due to ongoing research by Bataille et al.. The spatial analysis in R was done using the terra library and the Flux Footprint Prediction (FFP) model by Kljun et al. (2015). In Python, machine learning modeling was done with packages scikit-learn, xgboost and shap. Codes are available upon a reasonable request.

**Conclusion**

I think that the abstract clearly states the relevance of the study, also specifically related to governmental decisions: see text – "In the Netherlands, carbon dioxide (CO2) emissions from drained peatlands mount up to 5.6 Mton annually and, according the Dutch climate agreement, should be reduced by 1 Mton in 2030." I think the best way to end the story is to return in the conclusion a statement related to the governmental plans and the effects of changing groundwater levels. This would really make the story very "round" and would probably be an excellent ending of your manuscript. I was wondering in the end how much a measure, such as groundwater level changes, would help to achieve such planned reductions.

We added a final sentence, and agree that this makes the story more "round". "In conclusion, we have quantified the impact of groundwater changes on $CO_2$ fluxes across drained peatlands in the Netherlands, providing crucial understanding in support of the 1 Mton reduction target by the Dutch government."

**Detailed Comments**

**1) Abstract, Line 10**

"Using spatio-temporal data, we train and optimize a boosted regression tree (BRT) machine learning algorithm . . . " In this particular sentence it is not clearly mentioned what aim that training has. Please describe (short) what the BRT was trained to simulate.

We added that the model should predict instantaneous $CO_2$ fluxes. "Using spatio-temporal data, we train and optimize a boosted regression tree (BRT) machine learning algorithm to predict immediate $CO_2$ fluxes, (...)."

**2) Abstract, Lines 11-12**

"We find that emissions increase with 4.6 tonnes CO2 ha-1 yr-1 (90% CI: 4.0-5.4) for every 10 cm WTDe up to a WTDe of 0.8 meter." Here, the authors formulate the water table depth as a positive number (both for the indicated depth of 0.8 m. and also for the 10 cm increment). I assume that you are talking about the water table depth below the surface (e.g., -0.8 m) and about an increase in emissions with an increase (negative) of every 10 cm in water table depth. I suggest to make sure that this is clear to all readers when reading the abstract.

We changed the sentence to: "We find that emissions increase with 4.6 tonnes $CO_2$ ha$^{-1}$ yr$^{-1}$ (90% CI: 4.0-5.4) for every 10 cm lowering of the water table, down to a water table depth of 0.8 meter below the surface."

**3) Abstract, Lines 12-13**

"For more drained conditions, emissions decrease again, following an optimum-based curves". I am not really sure what the authors mean with "optimum-based curves" and would suggest to maybe use another wording here, to make sure that all readers understand what this refers to (forgive me if I am wrong, but I think it's not a widely recognized or standard term). Alternative, the authors could refer to a paper that describes this concept/ wording clearly.

With "optimum-based curve" we refer to the parabolic function that was fitted on the SHAP values (see Figs. 8 and 11), which has a maximum, or "optimum". However, the term "optimum-based" is not required for the reader's understanding, and we recognize that it may cause confusion. Hence, we omitted the last part of the sentence: "following an optimum-based curve."

**4) Abstract, Lines 13-14**

"Furthermore, we find that this effect is stronger in winter than in summer and that it varies between sites." Does this refer to an increase/decrease in fluxes with an increasing/decreasing water table depth? This was not fully clear to me.

Yes, it does indeed refer to the effect of WTD changes on $CO_2$ flux changes. We clarified this sentence: "Furthermore, we find that the sensitivity of $CO_2$ emissions to drainage is stronger in winter than in summer (...)."

**5) Abstract, Line 14**

"This study shows the added value of using ML..." ML Abbreviation was not introduced before, please define it here.

We added "(ML)" in line 7, where we first name machine learning.

**6) Reference(s) to Klimaatakkoord, 2019**

In the manuscript, the authors refer to a Dutch website where information is provided about the Dutch climate agreement (klimaatakkoord). I had difficulties finding the information here that was mentioned in the paper and would therefore suggest to directly cite the relevant document (https://www.klimaatakkoord.nl/binaries/klimaatakkoord/docum. Moreover, I would suggest to cite the English version of the document, as this is probably more relevant for the international community (https://www.government.nl/binaries/government/documenten/reports/2019/06/28/climate-agreement/Climate+Agreement.pdf). Finally, I assume it would be good to cite using the standard format (authors/organization - Government of the Netherlands) here.

Thank you for the comment and for linking the English pdf (!). We changed the reference, and are now citing as Government of the Netherlands (2019).

**7) Methods, comparison of airborne NEE and tower NEE fluxes for "Friesland"**

I generally really like this comparison and like this approach, but I think the following points are important to also mention in the manuscript.

(a) As far as I understand, the tower fluxes and airborne fluxes are not calculated in the same way. Airborne fluxes were calculated using the wavelet approach and tower data were calculated probably using a Reynolds

decomposition, correct? If this is the case, this probably should be stressed that this is another reason why direct intercomparison is not fully possible.

(b) You showed an example for Friesland where the fluxes have similar magnitudes. It would be great if the authors could mention that this was also the case for the other regions.

(c) The authors use a circular footprint for the footprints of the towers, and do not use the Kljun et al., (2015) model here. I understand the reason for doing so, but are these circular footprints homogeneous (which is what I would assume and what would "justify" the use of such footprints)?

**a)** It is true that fluxes were calculated differently, tower fluxes using Reynoulds decomposition and airborne fluxes using wavelet decomposition. While this difference should not have an impact on the fluxes because all flights were done during clear daytime conditions, it is nevertheless a difference and we added it as such in the description of Figure 4 (see point 14). In addition, to provide an overview of the different tower vs. airborne processing steps, we added a table in the Appendix, which we show here at point 11.

**b)** See point 14.

**c)** Thank you for pointing this out. Using circular footprints is clearly a shortcoming of our approach, and we suggest taking a wind-direction based average footprint in the future. We added this suggestion to the recommendations in Section 4.5.2. Still, within the scope and scale of the current study, the areas in the footprints are homogeneous, especially for the final selected features. In some cases, the footprint includes ditches, but they represent only a small fraction and are generally of minor importance for $CO_2$ fluxes.

**8) Methods, Line 91-93**

"A preliminary analysis we conducted based on a subset of the data with this method showed promising results for combining airborne and tower data, corresponding to existing estimates ". Here, I was not 100% sure to which existing estimates this sentence refers to? Do the authors refer to existing estimate of changes in CO2 emissions related to X cm change in groundwater levels in peatlands here (existing literature)? I suggest to specify what estimates are meant here, plus perhaps also including references to the existing literature.

Thank you for the comment. Here, "a preliminary analysis" refers to previous, unpublished work by our group with data only from the Groene Hart. Although this preliminary study served as a motivation for us to carry out the current study including also Friesland and Overijssel, we recognize that this information is not relevant to the wider public. We removed the sentence.

We also mention this preliminary study in the discussion, because of contrasting results. Here, we think it is relevant to mention the study, so we provided some clarification (Line 383): "In a preliminary, unpublished study we conducted using only the Groene Hart data, the addition of airborne data significantly improved the ML model ($R^2$ increased from 0.47 to 0.61)."

**9) Figure 2**

I am assuming that you are using the straight lines of the flight tracks for airborne EC flux calculations, and not the turns, correct? I suggest that it would be great if the figure would just show the tracks that were used for calculating fluxes (other regions could also show up, but maybe with an indication that these were not used for the flux calculation). I also would suggest to add a scalebar to the figure, to show the extend of the regions more clearly. Additionally, it would have actually been very nice to see how the peatlands are distributed over the

regions. Could the authors add peatland information to the (same) map or add another window that shows this additionally?

Following your suggestion we added the peat distribution as well as a scale bar. The pilots were instructed to keep the banking angle below 15 degrees during turns, to allow for flux calculations throughout the entire flight track (including the turns). Here, we show the updated figure and caption in Fig. 2.

[Figure]

Figure 2: The EC tower network used in this study, with three flight tracks over the study areas: **a)** Friesland, **b)** Overijssel and **c)** Groene Hart. Peat distribution is shown in orange. Airborne fluxes are calculated over the entire flight tracks, including the turn, since the banking angle was kept less than 15 degrees. General information on the EC sites and processing can be found in 2.1.3 and Appendix A. For more detailed, site-specific information, see Kruijt et al. (2023). The map was made in QGIS using an ESRI base map; peat distribution was obtained from the soil map (Wageningen Environmental Research, 2024).

**10) 2.1.2 Airborne Flux Measurements**

In the description of the airborne measurements (2.1.2 Airborne Flux Measurements), no explicit information about the sensors is given directly in this part of the section, which is important for comparability/ quality assessment etc. Only at the end of the section, after providing information about the airborne EC processing, a reference to Vellinga et al., (2013) is made: "For more detail on the aircraft and its equipment, see Vellinga et al. (2013)." I propose to

refer to this publication directly after the full description of the aircraft and its equipment is given.

We moved the sentence to line 122.

**11) Methods, Line 131**

"Further processing was done by following the framework of Foken et al. (2004)." Here, it is very unclear what further processing is done and one would need to look at the reference of Foken what specific processing step(s) the authors refer to. I assume this refers to QAQC analyses and would suggest that the authors add this information in one or more sentences to make this clear.

We provided some extra information in this sentence: "Further processing including quality checks and u* filtering was done following the framework of Foken et al. (2004). In Appendix A we provide an overview of the applied steps." We show the new Appendix with the overview table on the next page.

Table 1: Processing steps for tower and airborne flux calculation and filtering. EddyPro was used for tower fluxes.

| Processing step | Tower | Airborne |
|---|---|---|
| | Flux calculation | |
| Block-averaging | ✔(30 min) | ✔(2 km) |
| Wavelet decomposition | - | ✔ |
| Reynolds decomposition | ✔ | - |
| WPL[1] correction | ✔ | ✔ |
| Frequency loss correction | ✔ | High freq. not needed because at operating altitude flxues are carried by eddies <10Hz or 4m. Low freq. not needed because of wavelet decomposition. |
| Tilt correction radiation sensors[1] | - | ✔ |
| | Filters | |
| Signal strength filtering | ✔Remove Received Signal Strength Indicator (RSSI) < 70 (McDermitt et al., 2011) | - |
| Precipitation | ✔Remove timesteps with precipitation (sensor performance affected) | (no flights with precipitation) |
| Wind direction | ✔Exclude fluxes from undesired wind sectors (site-specific) | - |
| u* filtering | ✔Remove stable night data below u* threshold (site-specific, moving point test by Papale et al. (2006)) | |
| Stationarity and ITC[3] test | ✔ | ✔ |
| Meteorological measurements: physical range filter | ✔ | ✔ |
| Flux magnitude | $-100 < CO_2$ ($\mu$mol m$^{-2}$s$^{-1}$) $< 100$ | $-50 < CO_2$ ($\mu$mol m$^{-2}$s$^{-1}$) $< 50$ |
| Quality flags | ✔All hard flags by Vickers and Mahrt (1997). ✔Keep only quality flag = 2 (Foken et al., 2004). | ✔Keep quality flags for $CO_2$ and u* < 6 (Vellinga et al., 2013). |

[1] Webb, Pearman,& Leuning (Webb et al., 1980).
[2] Based on aircraft attitude and solar position.
[3] Integral Turbulence Characteristics

**12) Figure 3**

I like the idea of the image to show a typical flight leg over a region, and it demonstrates how you use five overlapping footprints for a 2km region. I also like the relationship to a tower and its footprint. I would, however, improve the text in the caption to make the connection to the manuscript text clearer. I also find the figure in its current state quite busy (a lot of lines and circles) and therefore more difficult to "read". I am wondering if one could simplify the image and for example leave out the orange lines with the distances between the footprint's maximum extend and the flight path. One could probably further simplify the footprints by only showing the 80% (?) footprint for each 2 km section (the 5 overlapping footprints) in general, and only show more detail when zooming in.

Other suggestions for improvements to the figure include to (1) again add a scale bar, (2) put either a legend with the meaning of the symbols and color (footprint) next to the figure or add information about the elements that are incorporated and their meaning in more detail to the caption, (3) add a wind rose to the image with average wind direction information for the flight.

I, assume that the footprint for the tower (ZEG_PT) is more of a yearly footprint and is not related to the particular wind direction of the "typical flight day" that you are showing here. It's also important to either adjust the figure to make sure that all footprints are specific for that particular example flight (also for the tower data), or I suggest to mention this in a note in the caption.

Thank you for the comment and suggestions. We adjusted the figure and caption:

[Figure]

Figure 3: Airborne sub-footprints of a typical flight over the Groene Hart, with flight altitude of 60 meters. In this study, we used static circular footprints for towers, which are shown in blue for LAW and ZEG_PT. In **a)**, five sub-footprints are visible for every 2-km window, where the contour lines represent the area from which 80% of the measured flux originates. All sub-footprints were overlaid with spatial data, and subsequently combined and normalized to get the final footprint values. The wind-rose shows the average wind direction. In **b)**, we show the contribution distribution within the first five sub-footprints: blue indicates the highest contribution, red indicates the lowest. The black 'x' denotes the ZEG_PT tower location. In both **a)** and **b)**, the differences in tower and airborne footprints are visualized.

**13) Section 2.1.3. EC Towers**

I was wondering if the towers that are used in this study are very similar or different in terms of sensor setup and if the processing of the data was all done in the same way (would be at least good to mention here). I assume they are not fully processed in the same way (one is done via wavelets, other probably using Reynolds decomposition?). Additionally, I think it is important to also mention here how the fluxes are calculated (at least mention how the half-hourly tower data was derived from the raw data). This does not need to be very lengthy, but int would at least be good to mention the software and perhaps some QAQC (e.g. EddyPro, REddyProc, TK3, your own processing pipeline), so one can compare the processing of tower and aircraft data easier without having to look at the referred report.

Thank you for the comment. Although all tower data is processed the same way, there are differences with airborne data processing. We recognize this was insufficiently clarified, and now added a few sentences in Section 2.1.3, as well as a comparison Table in the Appendix, which we showed at point 11.

"All sites were equipped with open-path gas analyzers, except LAW_MS_ICOS, which used a closed-path $CO_2$ sensor. Otherwise, sensor set ups were identical. The equipment was mounted on top of telescopic masts and arranged perpendicular to prevailing southwest winds. Measuring height ranged between 1.5 m and 6 m, based on the desired footprint size. Half-hourly fluxes were calculated using EddyPro LI-COR Biosciences (2023) and subsequently post-processed using a series of filters (see Appendix A). All processing was streamlined across towers, and no attempts at gap filling were made."

**14) Figure 4**

I personally find the figure a little bit small and not that easy to read. I would therefore propose to increase the size of all the text in the figure. I already think the figure clearly shows the point you are trying to make (that the airborne fluxes and tower fluxes have similar ranges). However, I would suggest to use distinct coloring between tower (one group of colors) and airborne data (a quite distinct and different color), so that it is even easier to see in the graph what the airborne data is showing and what the tower data is showing. One could potentially also add an n below each month showing how much data is available for that particular month.

We changed the colouring of the figure and added a kde plot for reference. Also, we extended thie figure caption.

[Figure]

Figure 4: **a)** Airborne and 'daytime' tower measurements between 10:00 and 16:00 of $CO_2$ fluxes in Friesland, binned per month. Boxes represent interquartile ranges, whiskers show minimum and maximum values, excluding outliers. Only a sub-period of the entire dataset is shown. **b)** Probability distributions of airborne and tower data, for all areas combined. Here, we use the same time frame for 'daytime' as in **a**. Although direct inter-comparison of airborne and tower measured fluxes is impossible due to intrinsic differences in footprints (see also Fig. 3), here we see similar magnitude and seasonality. Furthermore, fluxes are calculated differently: airborne using wavelet decomposition and tower using Reynolds decomposition. The airborne and tower fluxes show similar resemblance in the Groene Hart and Overijssel (not shown here).

**15) Table 2**

This table nicely shows the products that were used for the ML algorithm. I have two small remarks related to this table. (1) Can you provide information about the uncertainty of the variables described with these products (e.g. uncertainty in provided water table depth, height, peat depth, etc.)? This would be something that could be worthwhile to mention as I assume that these supplementary data sources also have info sheets where they provide information about the quality and uncertainty of their products. This would be good to add here. (2) Can you provide official references to the products that you have used and described in this table (often products come with a clear data citation)? Partly, these are provided in the text, but also not completely.

Thank you for the comment. For the maps where information on uncertainty could be found, we added the information to the table. We also added scientific references to the sources. The table is now as follows:

Table 2: Overview of supplementary data sources. For comparison, tower heights ranged between 1.5 and 6 meters, with footprints spanning several hundred meters; airborne flying altitude is 60 meters, with footprints spanning several kilometers (also see Fig. 3).

| Variable | Spatial res. | Temporal res. | Source |
|---|---|---|---|
| Land use* | 5 m | - | Landelijk Grondgebruik Nederland, LGN2020 (Hazeu et al., 2023) |
| Soil* | 5 m | - | Bodem Data, BOFEK2020 (Wageningen Environmental Research, 2024) |
| Elevation ($\pm$5 cm, Actueel Hoogtebestand Nederland (2025)) | 25 m | - | Algemeen Hoogtebestand Nederland, AHN3 (Rijkswaterstaat, 2019) |
| Peat depth ($\pm$ 10-30 cm, Wageningen Environmental Research (2015)) | 100 m | - | Bodem Data (Brouwer et al., 2023) |
| Groundwater level Air-filled pore space Soil moisture | 250 m | Daily | OWASIS product from Hydrologic, personal communication (retrieved January 9, 2024) |
| NDVI ($\pm$0.025, Didan (2025)) EVI ($\pm$ 0.025, Didan (2025)) | 250 m | 8 days | MODIS: MOD13Q1 Terra and Aqua (Didan, 2021) |

**16) Appendix A2**

Reclassification soil classes – the classes are provided here in Dutch and are probably not readable for an international audience. I would definitely add an English nomenclature connected to the Dutch names in the "New class" column. Both tables have a caption with the text "Figure A1 and Figure A2". I propose to change these captions to Table A1 and Table A2.

Thank you for the comment. We translated all "new class" names to English and changed the captions to Tables instead of Figures.

**17) Methods, Lines 188 – 189**

"Using the collected information, some additional covariates were calculated, such as effective water table depth (WTDe) based on groundwater level and elevation, the percentage of all peat classes together present in the footprint (AllPeat), peat on sand, and peat on peat. Combining peat depth with WTDe, the peat exposed to air ('exposed peat depth') in cm was calculated." This is very descriptive and does not provide the equations used in this particular case. Either provide a reference to the used equations or write down the equations in this part of the manuscript, so that it's fully clear how these covariates are calculated.

Thank you for the comment. As these 'calculations' are quite simple, it did not occur to us to provide formulas, but we understand that it enhances the reader's understanding. We added a few words and three formulas:

"Using the collected information, some additional covariates were calculated, such as effective water table depth (WTDe) based on groundwater level and elevation (Eq. 1), the percentage of all peat classes together present in the footprint (AllPeat) as well as for all peat on sand and peat on peat classes (Eq. 2). Combining peat depth with WTDe, the peat exposed to air ('exposed peat depth') in cm was calculated (Eq. 3).

$$WTD_e = elev. - OWASIS\_GW \tag{1}$$

$$AllPeat = \sum_{i=1}^{n} peatclass_i \tag{2}$$

$$ExpPeatDepth = min(PeatDepth, WTD_e) \tag{3}$$

where WTDe is effective water table depth below surface level; elev. is elevation; OWASIS_GW is groundwater level below sea level; peat class represents the fraction of the footprint in peat soil class $i$; ExpPeatDepth is exposed peat depth, or peat exposed to air. Eq. 2 was also used summing only peat on peat classes, and peat on sand classes (see Appendix A2 for soil classes)."

**18) Methods, Line 198**

"and by selecting one every four weeks of tower data," Please consider rephrasing this segment of the sentence.

We rephrased the sentence to the following: "For the airborne data, we created the test set by randomly selecting individual flight-legs. For the tower data, we divided the data in blocks of four weeks, using the first three for training and the fourth for testing."

**19) Methods, 2.3, Line 225**

"As we assume the underlying processes are the same, . . . " I think it is important that the authors explain here what processes they refer to (I assume to the physical processes in the peatland regions that steer the NEE fluxes?).

Indeed, we are referring to these physical processes. We added a few clarifying words: "As we assume the underlying physical processes steering NEE fluxes are the same, (...)."

**20) Methods, 2.3, Line 226**

"However, we expect the Overijssel model to be different, because while the aircraft covers agricultural land, we do not have any agricultural tower sites in that area, such as in the Groene Hart and in Friesland." Please consider rephrasing the segment "while the aircraft covers agricultural land" and refer to the aircraft flux data instead. I was also wondering why the model would be different if it still includes agricultural fluxes from aircraft data. Can the authors elaborate why this would be different from having similar fluxes from towers with small footprints? Are the aircraft footprints too large to capture a purely "agricultural sites" signal? Do these fluxes provide a less specific groundwater signal?

It is correct that the Overijssel model contains fluxes from agricultural areas because of the aircraft, but this is only a small part of the entire dataset: the Overijssel dataset consists of 2600 airborne measurements and 15000 tower measurements (Table 1), the latter all from natural areas. We rephrased Line 226 to the following:

"However, we expect the Overijssel model to be different, as all tower measurements in this region are from natural areas, unlike in the Groene Hart and Friesland, where sites are located on agricultural land. Although the Overijssel aircraft flux data covers agricultural land, the amount of airborne data points is limited compared to the tower data (see Table 1)."

**21) Methods, Sections 2.2 – 2.3**

I was a little bit lost in which features were used in the ML model, especially since section 2.3 already talks about the gapfilling of some of the features (see below), but the text does not go into detail on the features that were considered/ obtained after the tuning. While reading further, I understand that this information is provided rater in the results, but perhaps a little reference to the considered/ used features in this part of the text would also be nice and helpful for the reader (what features were considered in the beginning before the tuning).

We agree that a clarification at this earlier stage is beneficial. We added this on line 214: "The finally selected features are PAR (photosynthetically active radiation), Tsfc (surface temperature), RH (relative humidity), EVI (enhanced vegetation index) and WTDe (effective water table depth) and PeatDepth."

**22) Methods, Lines 241 – 243**

"As our sites contained gaps in meteorological data, we used publicly available hourly data from Dutch Meteorological Institution (KNMI) and interpolated in time to obtain half-hourly values. We used station Cabauw for the Groene Hart area and station Hoogeveen for the Overijssel and Friesland areas." Here, it would be important to mention which features you were gapfilling for your ML model (which meteo features did you consider). Please specify. Plus, do you talk here about the use of one "average" temperature, humidity etc. for each of the three regions (and no detailed spatial product - which is what I understood from your text)? Why did you not gapfill the meteorological data from the specific sites? In the following segment "interpolated in time to obtain" I assume you refer to a linear interpolation, correct? Please add the information about the interpolation method. On top of that,

as mentioned in the comment before this segment comes a little bit "out of the blue" as the features that you are using in your model are not fully described in the methods section 2.2 before, but appear more prominent in the results.

Thank you for the comment. We only used meteo data from the KNMI stations that was used in our final model, i.e. PAR, temperature and relative humidity. It is true that we only use two weather stations to represent all of our sites (Cabauw for the Groene Hart and Hoogeveen for Friesland and Overijssel). However, we do not consider this a significant issue. In general, these variables exhibit only small variation across the regions in the Netherlands. Furthermore, investigating spatial differences of meteorological variables was not the aim of the simulation exercise: we were interested in water fluctuations in space and time, as well as in seasonality processes, for which the data from the two KNMI stations are sufficient. We extended the paragraph as follows, also considering comment 23:

"We assembled half-hourly input data (values for PAR, Tsfc, RH, EVI, WTDe and PeatDepth) for each site over the years 2020, 2021 and 2022 from the various data sources (see Table 2). As our sites contained gaps in meteorological data (PAR, Tsfc, RH), we used publicly available hourly data from Dutch Meteorological Institution (KNMI) and linearly interpolated in time to obtain half-hourly values. By using the available data directly, we prevented inconsistencies in the time series that could arise with gap filling. We used KNMI stations "Cabauw" for the Groene Hart area and "Hoogeveen" for the Overijssel and Friesland areas, assuming that this limited spatial variability of meteorological variables is adequate for our sites. For the dynamic maps, we extracted the values at site locations for each day across the three years. Subsequently, using this continuous dataset of features, we let the model predict every half-hour $CO_2$ flux at every site, as well as under hypothetical scenarios where the WTDe was altered by $\pm 10$ cm. These predictions were then aggregated to construct annual NEE balances."

**23) Methods, Section 2.3 and last sentence**

"We let the model predict every half-hour flux and constructed annual NEE balances based on the actual WTDe level, as well as for hypothetical situations where the WTDe is altered by $\pm 10$ cm." In this section and in this segment, it is not clear where you are predicting your half-hourly NEE fluxes for. I assume you predict an average/ regional flux for all 3 regions and for all areas together? It would be very important to clearly specify this here in this part of the manuscript, so that the readers directly get what you are doing.

We specified the simulation settings. See the paragraph at comment 22.

**24) Results, 3.1**

The section starts with "In this section we present the model optimization results." Perhaps this sentence could be elaborated a little (what model? ML-based peatland NEE model - to do what?), so that people that might not read the whole manuscript can jump to this section and directly understand what the manuscript is about.

We added a few sentences at the start of the paragraph: "In this study, we trained several machine learning models on airborne and ground-based Eddy Covariance data to predict NEE fluxes from peatlands in the Netherlands, aiming to improve our understanding of groundwater-$CO_2$ dynamics. As a first step, we optimized model features and hyperparameters, to achieve models that were both high-performing and parsimonious. Here, we present these model optimization results."

**25) Results, Line 254**

"For some soil classes such as pV and hV" Please directly mention in the text what these classes are, so that people can directly read further, without having to look at the Appendix of you paper.

We added the English nomenclature in brackets: "For some soil classes such as "pV" (clayey earthy peat soils) and "hV" (thin peaty earthy peat soils >120 cm deep) (...)"

**26) Abbreviations ML features**

In the manuscript, the features that were used in the ML model to predict the fluxes are often only mentioned as abbreviations. This is totally fine if all abbreviations are explained in the text, which is the case with a large part of the features (such as EVI, NDVI, WTDe), but not for others (PAR, RH etc.) Please be sure to have all of these explained in the manuscript. On top of that, I would suggest to give full names in the captions of relevant figures (for example Figure 6) and your Appendix (for example Appendix C, Figure C.1). Units (if relevant) for these features would also be good to clearly mention here.

Thank you for the comment and suggestion. We added units and explanation on abbreviations in Figure 6, and ensured all abbreviations were explained in the text. In the caption of Figure C.1 we explained all abbreviations found in the figure.

**27) Appendix C, Figure C.1**

I suggest to make this figure larger in the Appendix to make it easier to read. Also, as mentioned before, I would explain the features in the caption, so that it's clear to the reader what features they are looking at, without having to search for their meaning elsewhere. I also noticed that some features do not show up for a particular case (airborne/ tower/ merged) and assume this is because of their low correlation score in that particular case. Is that true? If that's the case, it would be good to mention in the caption.

We enlarged the figure. The sub-figures of Figure C.1 all show the same subset of features, which was selected based on overall performance of features. All meteorological, water- and peat-related features were included in this subset, as well as some soil and land use classes. We do not show the (very low) correlations for all soil and land use classes, as we deem this unnecessary.

We changed the figure caption to the following: "Pearson correlation between a selection of features and $CO_2$ flux for airborne, tower and merged datasets per area. The selected features are the most correlated throughout all datasets. The other features (for example, other soil classes and land use classes) are not shown here, for simplicity of the figure. Abbreviations of features: PAR_f: photosynthetic active radiation, RH: relative humidity, Tsfc: surface temperature, EVI: enhanced vegetation index, NDVI: normalized difference vegetation index, AllPeat: percentage of footprint in peat classes, hV: percentage of footprint in soil class thin peaty earthy peat soils (>120cm deep), pV: percentage of footprint in soil class clayey earthy peat soils, kV: soil class clayey peat soils (>120cm deep), Wat: percentage of footprint in land use class open water, Grs: percentage of footprint in land use class grasslands, OWASIS_BB: available open pore space from OWASIS, OWASIS_BV: soil moisture from OWASIS, OWASIS_GW: groundwater level from OWASIS (mbs), WTDe: effective water table depth, ExposedPeat: peat depth exposed to air."

**28) Table Appendix D1**

There is a comma (",") at the end of the caption. Please replace with a ".

Done.

**29) Table 3**

Based on this table's $R^2$ values (bef. vs. aft.), the hyperparameter tuning improved the model only a little bit for most regions. However, the improvement for the Overijssel region is really large. I think this was not really discussed in the manuscript, but it would actually be interesting to see why the hyper tuning improved the model that much for this particular region.

Thank you for the remark, it is true we had not touched upon that aspect. We added some lines after line 364: "Notably, only the Overijssel model was strongly improved by hyperparameter tuning. We hypothesize that the low initial performance scores are a result of the smaller dataset size and lower data quality. Increasing the model complexity (by lowering the learning rate and increasing the number of estimators) the model was able to identify relationships nonetheless. However, when attempting to generalize to other areas, it became clear the model was overfitting."

**30) Figure 10 and Results Section 3.3 ("Simulations")**

First of all, it would be good to mention here, where the "Range in literature" comes from (one or multiple citations). This range is also potentially not always very visible. Probably one could work with a transparency to make the boxplots in the front and the "range in literature" both visible. Perhaps in this case violin plots, instead of boxplots would also give a clearer indication of the distribution of the data.

We provided an explanation for 'Range in literature' in the figure caption, and made the boxplots narrow such that the grey band is clearly visible.

[Figure]

Figure 10: Monthly binned differences in NEE predictions when WTDe was lowered by 10 cm. The color represents the average temperature in our data for that specific month. The 'range in literature' is based on the lowest and highest estimates of the groundwater-$CO_2$ slopes in the literature: 2.1 t $CO_2$ ha$^{-1}$ yr$^{-1}$ per 10 cm as found by Kruijt et al. (2023) and 4.5 t $CO_2$ ha$^{-1}$ yr$^{-1}$ per 10 cm as found by Fritz et al. (2017). See Fig. 11 and Appendix G for other estimates).

**31) Figure 11**

The comparison with other studies is interesting, I think it is also important to think about the fact that these studies all identify different functions based on different datasets using partly different measurement methods and data originating from different areas (locations). This should probably also be shortly addressed in the discussion.

We added this on line 410: "In addition, they investigate different locations, partly use different measurement methods - chamber or tower - and they are based on multi-site comparisons thus indicate mostly spatial dependencies."

**32) 4.4 Implications for mitigation strategies**

I suggest to make a link here to the plans of the Dutch government and what role your study results play here.

Thank you for the comment. We illuminated the link between our study and actual policy plans by adding the following few lines to Section 4.4: "However, the evaluation of potential mitigation measures did not fall within the scope of our study. As we did not incorporate data on mitigation efforts into the model, we cannot draw conclusions in that regard. Nevertheless, this study is part of the Netherlands Research Programme on Greenhouse Gas Dynamics in Peatlands and Organic Soils (NOBV), and contributes to the corresponding measuring, monitoring and modelling framework. Within this framework, the efficiency of mitigation measures is widely studied, using both data-driven methods as well as process-based models such as SOMERS (Erkens et al., 2022) and findings will be reported to policy-makers."

**References**

Aben, R. C. H., Craats, D. V. D., Boonman, J., Peeters, S. H., Vriend, B., Boonman, C. C. F., Velde, Y. V. D., Erkens, G., & Berg, M. V. D. (2024). Using automated transparent chambers to quantify CO2 emissions and potential emission reduction by water infiltration systems in drained coastal peatlands in the Netherlands. *EGUsphere*, *403*. https://doi.org/10.5194/egusphere-2024-403

Actueel Hoogtebestand Nederland. (2025). Kwaliteitsbeschrijving [Accessed: 2025-04-14]. https://www.ahn.nl/kwaliteitsbeschrijving

Brouwer, F., Assinck, F., Harkema, T., Teuling, K., & Walvoort, D. (2023). *Actualisatie van de bodemkaart in de gemeente vijfheerenlanden: Herkartering van de verbreiding van veen* (tech. rep.). WOT Natuur & Milieu.

Didan, K. (2021). Mod13q1 modis/terra vegetation indices 16-day l3 global 250m sin grid v061 [Accessed: 2025-04-14]. https://doi.org/10.5067/MODIS/MOD13Q1.061

Didan, K. (2025). Modis vegetation indices (mod13) validation status [Accessed: 2025-04-14]. https://modis-land.gsfc.nasa.gov/ValStatus.php?ProductID=MOD13

Erkens, G., Melman, R., Jansen, S., Boonman, J., van der Velde, Y., Hefting, M., Keuskamp, J., van den Berg, M., van den Akker, J., Fritz, C., Bootsma, H., Aben, R., Hessel, R., Hutjes, R., van Asselen, S., Harpenslager, S. F., Kruijt, B., & the NOBV consortium. (2022). Somers: Monitoring greenhouse gas emission from the dutch peatland meadows on parcel level. *EGU General Assembly 2022 Proceedings*. https://doi.org/10.5194/egusphere-egu22-12177

Foken, T., Göockede, M., Mauder, M., Mahrt, L., Amiro, B., & Munger, W. (2004). Post-field data quality control. In *Handbook of micrometeorology: A guide for surface flux measurement and analysis* (pp. 181–208). Springer.

Fritz, C., Geurts, J., Weideveld, S., Temmink, R., Bosma, N., Wichern, F., Smolders, A., & Lamers, L. (2017). Meten is weten bij bodemdaling-mitigatie. effect van peilbeheer en teeltkeuze op co2-emissies en veenoxidatie. *Bodem*, 20.

Government of the Netherlands. (2019, June). Climate agreement [Accessed: 2025-04-05]. https://www.government. nl/binaries/government/documenten/reports/2019/06/28/climate-agreement/Climate+Agreement.pdf

Hazeu, G., Schuiling, R., Thomas, D., Vittek, M., Storm, M., & Bulens, J. D. (2023). *Landelijk grondgebruiksbestand nederland 2021 (lgn2021): Achtergronden, methodiek en validatie* (tech. rep.). Wageningen Environmental Research.

Kljun, N., Calanca, P., Rotach, M. W., & Schmid, H. P. (2015). A simple two-dimensional parameterisation for Flux Footprint Prediction (FFP). *Geosci. Model Dev.*, *8*(11), 3695–3713. https://doi.org/10.5194/GMD-8-3695-2015

Kruijt, B., Buzacott, A., van Giersbergen, Q., Bataille, L., Biermann, J., Berghuis, H., Heuts, T., Jans, W., Nouta, R., & van Huissteden Hefting F Hoogland, J. M. (2023). Co2 emissions from peatlands in the netherlands: Drivers of variability in eddy covariance fluxes.

LI-COR Biosciences. (2023). *Li-cor eddy covariance instrumentation and software user manual* [Accessed: 2025-04-11]. LI-COR Biosciences. https://licor.app.boxenterprise.net/s/1ium2zmwm6hl36yz9bu4

McDermitt, D., Burba, G., Xu, L., Anderson, T., Komissarov, A., Riensche, B., Schedlbauer, J., Starr, G., Zona, D., Oechel, W., et al. (2011). A new low-power, open-path instrument for measuring methane flux by eddy covariance. *Applied Physics B*, *102*, 391–405.

Papale, D., Reichstein, M., Aubinet, M., Canfora, E., Bernhofer, C., Kutsch, W., Longdoz, B., Rambal, S., Valentini, R., Vesala, T., & Yakir, D. (2006). Towards a standardized processing of net ecosystem exchange measured with eddy covariance technique: Algorithms and uncertainty estimation. *Biogeosciences*, *3*(4), 571–583.

Rijkswaterstaat. (2019). Actueel hoogtebestand nederland (ahn3) [Accessed: 2025-04-14]. https://www.pdok.nl/introductie/-/article/actueel-hoogtebestand-nederland-ahn

Vellinga, O. S., Dobosy, R. J., Dumas, E. J., Gioli, B., Elbers, J. A., & Hutjes, R. W. (2013). Calibration and Quality Assurance of Flux Observations from a Small Research Aircraft. *J. Atmos. Ocean. Technol.*, *30*(2), 161–181. https://doi.org/10.1175/JTECH-D-11-00138.1

Vickers, D., & Mahrt, L. (1997). Quality control and flux sampling problems for tower and aircraft data. *Journal of atmospheric and oceanic technology*, *14*(3), 512–526.

Wageningen Environmental Research. (2015). Veendikte 2014 [Accessed: 2025-04-14]. https://opendata.zuid-holland.nl/geonetwork/srv/api/records/098B74D3-D49B-422A-BCA4-6C11A3FA7D2A

Wageningen Environmental Research. (2024). Bodemkaart van nederland [Accessed: 2025-04-11]. https://www.bodemdata.nl

Webb, E. K., Pearman, G. I., & Leuning, R. (1980). Correction of flux measurements for density effects due to heat and water vapour transfer. *Quarterly Journal of the Royal Meteorological Society*, *106*(447), 85–100.